# ANALOGY-FORMING TRANSFORMERS FOR FEW-SHOT 3D PARSING

**Nikolaos Gkanatsios**\*, **Mayank Singh**\*, **Zhaoyuan Fang**,
**Shubham Tulsiani** & **Katerina Fragkiadaki**
School of Computer Science
Carnegie Mellon University
Pittsburgh, PA 15213, USA
{ngkanats,mayanks2,zhaoyuaf,stulsian,katef}@cs.cmu.edu

## ABSTRACT

We present Analogical Networks, a model that encodes domain knowledge explicitly, in a collection of structured labelled 3D scenes, in addition to implicitly, as model parameters, and segments 3D object scenes with analogical reasoning: instead of mapping a scene to part segments directly, our model first retrieves related scenes from memory and their corresponding part structures, and then predicts *analogous* part structures for the input scene via an end-to-end learnable modulation mechanism. By conditioning on more than one retrieved memories, compositions of structures are predicted, that mix and match parts across the retrieved memories. One-shot, few-shot or many-shot learning are treated uniformly in Analogical Networks, by conditioning on the appropriate set of memories, whether taken from a single, few or many memory exemplars, and inferring analogous parses. We show Analogical Networks are competitive with state-of-the-art 3D segmentation transformers in many-shot settings, and outperform them, as well as existing paradigms of meta-learning and few-shot learning, in few-shot settings. Analogical Networks successfully segment instances of novel object categories simply by expanding their memory, without any weight updates.

> Ask not what it is, ask what it is like.
>
> *Moshe Bar*

## 1 INTRODUCTION

The dominant paradigm in existing deep visual learning is to train high-capacity networks that map input observations to task-specific outputs. Despite their success across a plethora of tasks, these models struggle to perform well in *few-shot* settings where only a small set of examples are available for learning. Meta-learning approaches provide one promising solution to this by enabling efficient task-specific adaptation of generic models, but this specialization comes at the cost of poor performance on the original tasks as well as the need to adapt separate models for each novel task.

We introduce Analogical Networks, a semi-parametric learning framework for 3D scene parsing that pursues analogy-driven prediction: instead of mapping the input scene to part segments directly, the model reasons analogically and maps the input to modifications and compositions of past labelled visual experiences. Analogical Networks encode domain knowledge explicitly in a collection of structured labelled scene memories as well as implicitly, in model parameters. Given an input 3D scene, the model retrieves relevant memories and uses them to modulate inference and segment object parts in the input point cloud. During modulation, the input scene and the retrieved memories are jointly encoded and contextualized via cross-attention operations. The contextualized memory part features are then used to segment analogous parts in the 3D input scene, binding the predicted part structure to the one from memory, as shown in Figure 1. Given the same input scene, the

---

\*Equal contribution

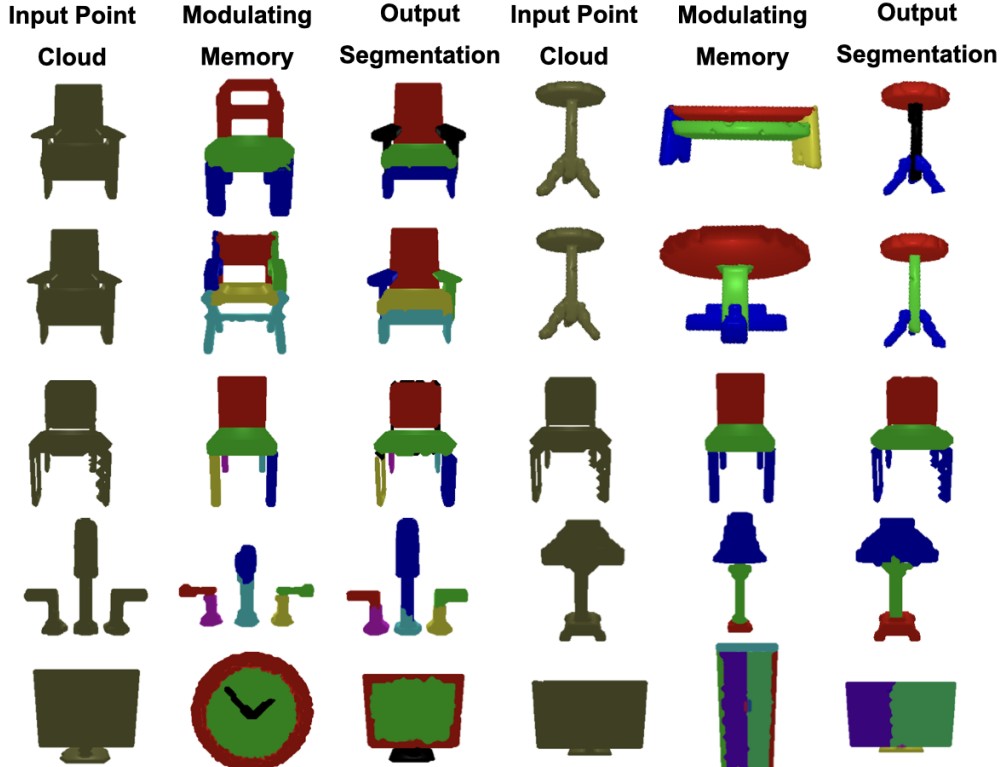

Figure 1: **Analogical Networks form analogies between retrieved memories and the input scene** by using memory part encodings as queries to localize corresponding parts in the scene. Retrieved memories (2nd and 5th columns) modulate segmentation of the input 3D point cloud (1st and 4th columns, respectively). We indicate corresponding parts between the memory and the input scene with the same color. Cross-object part correspondences emerge even without any part association or semantic part labelling supervision (5th row). For example, the model learns to correspond the parts of a clock and a TV set, without ever trained with such cross scene part correspondence. Parts shown in black in columns 3 and 6 are decoded from scene-agnostic queries and thus they are not in correspondence to any parts of the memory scene. Conditioning the same input point cloud on memories with finer or coarser labellings results in segmentation of analogous granularity (3rd row).

output prediction changes with varying conditioning memories. For example, conditioned on visual memories of varying label granularity, the model segments the input in a granularity analogous to the one of the retrieved memory. One-shot, few-shot or many-shot learning are treated uniformly in Analogical Networks, by conditioning on the appropriate set of memories. This is very beneficial over methods that specifically target few-shot only scenarios, since, at test time, an agent usually cannot know whether a scene is an example of many-shot or few-shot categories.

Analogical Networks learn to bind memory part features to input scene part segments. Fine-grained part correspondence annotations across two 3D scenes are not easily available. We devise a novel within-scene pre-training scheme to encourage correspondence learning across scenes. We augment (rotate and deform) a given scene in two distinct ways, and train the modulator to parse one of them given the other as its modulating memory, bypassing the retrieval process. During this within-scene training, we have access to the part correspondence between the memory and the input scene, and we use it to supervise the query-to-part assignment process. We show within-scene training helps our model learn to associate memory queries to similarly labelled parts *across scenes* without ever using cross-scene part correspondence annotations, as shown in Figure 1.

We test our model on the PartNet benchmark of Mo et al. (2019) for 3D object segmentation. We compare against state-of-the-art (SOTA) 3D object segmentors, as well as meta-learning and few-shot learning (Snell et al., 2017) baselines adapted for the task of 3D parsing. Our experiments

show that Analogical Networks perform similarly to the parametric alone baselines in the standard many-shot train-test split and particularly shine over the parametric baselines in the few-shot setting: Analogical Networks segment novel instances much better than parametric existing models, simply by expanding the memory repository with encodings of a few exemplars, even without any weight updates. We further compare against variants of our model that consider memory retrieval and attention without memory query binding, and thus lack explicit analogy formation, as well as other ablative versions of our model to quantify the contribution of the retriever and the proposed within-scene memory-augmented pre-training. Our code and models are publicly available in the project webpage: http://analogicalnets.github.io/.

## 2 RELATED WORK

**Few-shot prediction: meta-learning and learning associations**    A key goal for our approach is to enable accurate inference in few-shot settings. Previous approaches (Finn et al., 2017; Rusu et al., 2018; Snell et al., 2017; Wang et al., 2018b; Bar et al., 2022; Mangla et al., 2020; Wang et al., 2020; Nguyen & Todorovic, 2019; Liu et al., 2020; Tian et al., 2020; Yang et al., 2020) that target similar settings can be broadly categorized as relying on either meta-learning or learning better associations. Meta-learning approaches tackle few-shot prediction by learning a generic model that can be efficiently adapted to a new task of interest from a few labelled examples. While broadly applicable, these methods result in catastrophic forgetting of the original task during adaptation and thus require training a new model for each task of interest. Moreover, the goal of learning generic and rapidly adaptable models can lead to suboptimal performance over the base tasks with abundant data. An alternative approach for the few-shot setting is to learn better associations. For example, the category of a new example may be inferred by transferring the label(s) from the one (or few) closest samples (Snell et al., 2017; Sung et al., 2018). While this approach obviates the need for adapting models and can allow prediction in few-shot and many-shot settings, the current approaches are only applicable to global prediction, e.g., image labels. Our work can be viewed as extending such association-based methods to allow predicting fine-grained and generic visual structures using our proposed modulation-based prediction mechanism.

**Memory-augmented neural networks** Analogical Networks is a type of memory-augmented neural networks. Memory-augmentation of parametric models permits **fast learning** with few examples, where data are saved in and can be accessed from the memory immediately after their acquisition (Santoro et al., 2016), while learning via parameter update is slow and requires multiple gradient iterations on de-correlated examples. External memories have recently been used to scale up language models (Borgeaud et al., 2021; Khandelwal et al., 2019), and alleviate from the limited context window of parametric transformers (Wu et al., 2022), as well as to store knowledge in the form of entity mentions (de Jong et al., 2021), knowledge graphs (Das et al., 2022) and question-answer pairs (Chen et al., 2022). Memory attention layers in these models are used to influence the computation of transformer layers and have proven very successful for factual question answering, but also for sentence completion over their parametric counterparts.

**In-context learning**    In-context learning (ICL) (Brown et al., 2020) aims to induce a model to perform a task by feeding in input-output examples along with an unlabeled query example. The primary advantage of in-context learning is that it enables a single model to perform many tasks immediately without fine-tuning. Analogical Networks are in-context learners in that they infer the part segmentation of an object 3D point cloud in the context of retrieved labelled object 3D point clouds. While in language models ICL emerges at test time while training unsupervised (and out-of-context) for language completion, Analogical Networks are trained in-context with related examples using supervision and self-supervision. Although in prompted language models (Huang et al., 2022; Brown et al., 2020) the input-output pairs are currently primarily decided by the engineer, in Analogical Networks they are automatically inferred by the retriever.

## 3 ANALOGICAL NETWORKS FOR 3D OBJECT PARSING

The architecture of Analogical Networks is illustrated in Figure 2. Analogical Networks are analogy-forming transformer networks for part segmentation where queries that decode entities in the input scene are supplied by retrieved part-encoded labelled scenes, as well as by the standard set of

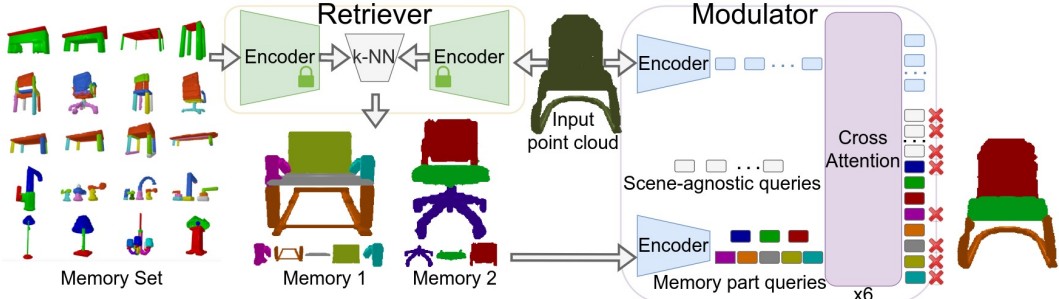

Figure 2: **Architecture for Analogical Networks.** Analogical Networks are comprised of retriever and modulator sub-networks. In the retriever, labelled memories and the (unlabelled) input point cloud are separately encoded into feature embeddings and the top-$k$ most similar memories to the present input are retrieved. In the modulator, each retrieved memory is encoded into a set of part feature embeddings and initializes a query that is akin to a slot to be "filled" with the analogous part entity in the present scene. These queries are appended to a set of learnable parametric scene-agnostic queries. The modulator contextualizes the queries with the input point cloud through iterative self and cross-attention operations that also update the point features of the input. When a memory part query decodes a part in the input point cloud, we say the two parts are put into correspondence by the model. We color them with the same color to visually indicate this correspondence.

scene-agnostic parametric queries of detection transformers (Carion et al., 2020). When a memory part query is used to decode a part segmentation in the input scene, we say the two parts, in the memory and the input, are put into correspondence. By using memory queries to decode parts in the input, our model forms analogies between detected part graphs in the input scene and part graphs in the modulating memories. Then, metadata, such as semantic labels, attached on memory queries automatically propagate to the detected parts.

Analogical Networks are comprised of two main modules: (i) A retriever, that takes as input a 3D object point cloud and a memory repository of labelled 3D object point clouds, and outputs a set of relevant memories for the scene at hand, and (ii) a modulator, that jointly encodes the memories and the input scene and predicts its 3D part segmentation.

**Retriever** The retriever has access to a memory repository of labelled 3D object point clouds. Each labelled training example is a memory in this repository. Examples labelled with different label granularities constitute separate memories. The retriever encodes each memory example as well as the input point cloud into distinct normalized 1D feature encodings. The top-$k$ memories are retrieved by computing an inner product between the input point cloud feature and the memory features.

**Modulator** The modulator takes as input the retrieved memories and the unlabelled input point cloud and predicts part segments and semantic labels for the present scene. The input scene is encoded into a set of 3D point features. Each memory scene is encoded into a set of 1D part encodings, one for each annotated part in the memory, which we call memory part queries $f^m \in \mathbb{R}^{c \times M}$, in accordance to parametric queries used to decode objects in DETR (Carion et al., 2020). The input points, memory part queries, and parametric queries are contextualized via a set of cross and self-attention operations, that iteratively update all queries and point features. Then, each of the contextualized queries predicts a segmentation mask proposal through inner-product with the contextualized point features, as well as an associated confidence score for the predicted part. These part segmentation proposals are matched to ground-truth part binary masks using Hungarian matching (Carion et al., 2020). For the mask proposals that are matched to a ground-truth mask, we compute the segmentation loss, which is a per-point binary cross-entropy loss (Vu et al., 2022; Cheng et al., 2021b). We also supervise the confidence score of all queries. For implementation details, please see the Appendix, Section 6.1.

Our modulator network resembles detection and segmentation transformers for 2D images (Carion et al., 2020; Cheng et al., 2021a) where a set of learnable 1D vectors, termed parametric queries, iteratively cross-attend to input image features and self-attend among themselves to predict object segmentation masks in the input scene. The key difference between our modulator network and

existing detection transformers are that retrieved memory entity encodings are used as queries, i.e., candidates for decoding parts in the input point cloud, alongside the standard set of scene-agnostic parametric queries, as shown in Figure 2. Additional differences are that we update the point features alongside the queries in the cross-attention layers, which we found helped performance. Finally, we operate in 3D point clouds as opposed to 2D images.

## 3.1 TRAINING

Analogical Networks aim to learn to associate parts in the retrieved memory graphs with parts in the input point cloud. Semantic part labels can only partially supervise this fine-grained part association, since retrieved memories can be too different from the input scene. We train our model in two stages to facilitate this fine-grained association learning:

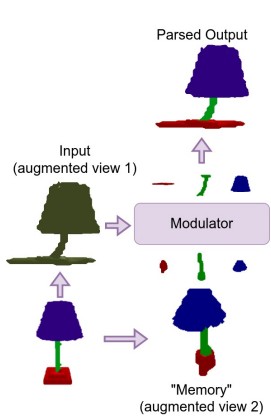

Figure 3: **Within-scene pre-training.**

**1. Within-scene training:** We apply two distinct augmentations (rotations and deformations (Kim et al., 2021)) to each training 3D point cloud, and use one as the input scene and the other as the modulating memory, as shown in Figure 3. In this case, we have access to ground-truth part associations between the parts of the memory and the input scene, which we use to supervise each memory part query to decode the corresponding part in the input cloud, and we do not use Hungarian matching. We call this within-scene training since both the input and memory come from the same scene instance.

**2. Cross-scene training:** During cross-scene training, the modulating memories are sampled from the top-$k$ retrieved memories per input scene. We show in our experimental section that accurate memory query to input part associations emerge during Hungarian matching in cross-scene training, even without using part association ground-truth, thanks to the within-scene pre-training. This is important, as often we cannot have fine-grained annotations of structure correspondence across scene exemplars.

In Analogical Networks, the retrieval process is not end-to-end differentiable with respect to the downstream scene parsing task. This is because i) we have no ground-truth annotations for retrieval, and ii) the retrieved memories are contextualized with the input through dense cross-attention operations which do not allow gradients to flow to the retriever's encoding process. We pre-train the encoder and modulator parameters independently of the retriever using within-scene training. We then use the modulator's encoder weights as the retriever's frozen encoder (adding a parameter-free average pooling layer on top). In the Appendix, we include pseudo-code for within-scene training in Algorithm 1 and cross-scene training in Algorithm 2.

## 4 EXPERIMENTS

We test Analogical Networks on PartNet (Mo et al., 2019), an established benchmark for 3D object segmentation. PartNet contains 3D object instances from multiple object categories, annotated with parts in three different levels of granularity. We split PartNet object categories into base and novel categories. Our base categories are Chair, Display, Storage Furniture, Bottle, Clock, Door, Earphone, Faucet, Knife, Lamp, Trash Can, Vase and our novel categories are Table, Bed, Dishwasher and Refrigerator. We consider two experimental paradigms: **1. Many-shot:** For the exemplars of the base categories we consider the standard PartNet train/test splits. Our model and baselines are trained in the base category training sets, and tested on segmenting instances of the base categories in the test set. **2. Few-shot:** $K$ labelled examples from each novel category are given and the model is tasked to segment new examples of these categories. This tests few-shot adaptation. We consider $K = 1$ and $K = 5$. We aggregate results across multiple $K$-shot learning tasks (episodes). We test two versions of our model: i) `AnalogicalNets single-mem`, which is Analogical Networks with a single modulating memory, and ii) `AnalogicalNets multi-mem`, with five modulating memories. Unless mentioned explicitly, Analogical Networks will imply the single-memory model.

Our experiments aim to answer the following questions:

1. How do Analogical Networks compare against parametric-alone state-of-the-art models in many-shot and few-shot 3D object segmentation?

2. How well do Analogical Networks adapt few-shot via memory expansion, without any weight update?

3. How do Analogical Networks compare against alternative memory-augmented networks where retrieved memory part encodings are attended to but not used for decoding parts?

4. How well do Analogical Networks learn part-based associations across scenes without part association ground-truth and semantic label supervision?

**Evaluation metrics**   We use the Adjusted Random Index (ARI) as our label-agnostic segmentation quality metric (Rand, 1971), which is a clustering score ranging from $-1$ (worst) to 1 (best). ARI calculates the similarity between two point clusterings while being invariant to the order of the cluster centers. We compute $100\times$ ARI using the publicly available implementation of (Kabra et al., 2019). We use mean Average Precision (mAP) per part (Mo et al., 2019) for semantic part instance segmentation and mean intersection over union (mIoU) for 3D point semantic segmentation.

**Baselines**   We compare our model to existing models in the literature. We further compare against strong parametric baseline models we develop; the latter ended up outperforming all previous existing works in the many-shot settings. We consider the following baseline models:

- `PartNet` of (Mo et al., 2019) is a 3D segmentation network with the same backbone as our model. This model implements "queries" as a fixed number of sets of MLPs that operate over the encoded input point cloud. Each set contains one MLP for per-pixel confidence and one for part confidence. The same losses used by our model are used to supervise this baseline.

- `DETR3D` is a 3D segmentation transformer network with the same backbone as our model and similar segmentation prediction heads and losses, but without any memory retrieval, akin to the 3D equivalent of a state-of-the-art 2D image segmentor (Carion et al., 2020; Athar et al., 2021). We update the point features in the decoder layers, same as in our model, which we found to boost performance. Contrary to `DETR3D`, Analogical Networks attend to external memories.

- `PrototypicalNets` is an adaptation of the episodic prototypical networks for image classification of (Snell et al., 2017) to the task of 3D object part segmentation. Specifically, given a set of N labelled point clouds, we form average feature vectors for each semantic labelled part and use them as queries to segment points into corresponding part masks through inner-product decoding. Contrary to `PrototypicalNets`, Analogical Networks contextualize the memory queries and the input scene. `PrototypicalNets` require a retriever that knows the category label of the input scene, so it has access to privileged information. Without this assumption, we got very low performance from this model.

**Ablations**   We compare our model to the following variants and ablative versions:

- `Re-DETR3D` (Retrieval-DETR3D) is a variant of Analogical Networks that attends to retrieved part memory encodings but does not use them to decode parts. Instead, all parts are decoded from the parametric scene-agnostic queries. Different from Analogical Networks, analogies cannot emerge between a memory and the input scene since this model does not represent such correspondence explicitly, but only implicitly, in the attention operations.

- `AnalogicalNets w/o pretrain` is our `AnalogicalNets single-mem` model without any within-scene (pre)training.

- `AnalogicalNets OrclCategoryRetrv` is our `AnalogicalNets single-mem` model with a privileged retriever that has access to the ground-truth category label of the input and only retrieves memories of the same category label.

## 4.1 MANY AND FEW-SHOT 3D OBJECT SEGMENTATION

The PartNet benchmark provides three levels of segmentation annotations per object instance where level 3 is the most fine-grained. We train and test our model and baselines on all three levels. We use a learnable level embedding as additional input for our baselines `PartNet` and `DETR3D`, as is

| Method | Fine-tuned? | Novel Categories | | Base Categories |
| --- | --- | --- | --- | --- |
| | | 1-shot ARI (↑) | 5-shot ARI (↑) | Many shot ARI (↑) |
| `PartNet` | ✗ | 26.1 | 26.1 | 54.3 |
| | ✓ | $22.0 \pm 0.90$ | $25.9 \pm 0.87$ | - |
| `DETR3D` | ✗ | 30.4 | 30.4 | **74.3** |
| | ✓ | $39.4 \pm 1.44$ | $52.7 \pm 1.44$ | - |
| `Re-DETR3D` | ✗ | $38.2 \pm 1.79$ | $46.8 \pm 0.66$ | **74.3** |
| | ✓ | $46.5 \pm 2.61$ | $55.6 \pm 1.82$ | - |
| `AnalogicalNets single-mem` | ✗ | $\mathbf{49.0 \pm 0.80}$ | $52.0 \pm 1.11$ | 72.5 |
| | ✓ | $\mathbf{50.6 \pm 2.72}$ | $\mathbf{57.0 \pm 1.33}$ | - |
| `AnalogicalNets multi-mem` | ✗ | - | $\mathbf{52.1 \pm 0.75}$ | 74.2 |
| | ✓ | - | $56.4 \pm 1.81$ | - |
| `AnalogicalNets OrclCategoryRetrv` | ✗ | $51.2 \pm 0.96$ | $53.8 \pm 1.03$ | 75.6 |
| | ✓ | $52.6 \pm 2.96$ | $58.3 \pm 1.36$ | - |

Table 1: **Semantics-free 3D part segmentation performance** on the PartNet benchmark. Without any fine-tuning, Analogical Networks outperform `DETR3D` by more than 20% in the few-shot setup. Even upon fine-tuning, Analogical Networks outperform `DETR3D` by 4.3% ARI.

usually the case in multi-task models (Jang et al., 2022). In the many-shot setting, we train our model and baselines jointly across all base categories and test them across all of them as well, using the standard PartNet train/test splits. For Analogical Networks and `Re-DETR3D`, all examples in the training set become part of their memory repository. In the few-shot setting, `PartNet` and `DETR3D` adapt by weight finetuning on the $K$-shot task. Analogical Networks and `Re-DETR3D` adapt in two ways: i) by expanding the memory of the model with the novel $K$-shot support examples, and ii) by further adapting the weights via fine-tuning to segment the $K$ examples. In this case, the memory set is only the novel labelled support set instances. **The retriever does not have access to the object category information in any of the many-shot or few-shot settings** unless explicitly stated so.

### 4.1.1 SEMANTICS-FREE INSTANCE SEGMENTATION

In this section, we train *all models and baselines for semantics-free object part segmentation, without any semantic labelling objectives*. We show quantitative semantic-agnostic many-shot and few-shot part segmentation results in Table 1. For the few-shot setting, we show both fine-tuned and non-fine-tuned models. For the few-shot performance, we report mean and standard deviation over 10 tasks where we vary the $K$-shot support set. Our conclusions are as follows:

**(i)** Analogical Networks dramatically outperform `DETR3D` in few-shot part segmentation. While in the many-shot setting the two models have similar performance, when adapting few-shot to novel categories, Analogical Networks and all their variants dramatically outperform parametric alone `DETR3D`, both before and after fine-tuning.

**(ii)** Analogical networks can adapt few-shot simply by memory expansion, without weight updates. Indeed, the 5-shot performance of our model is close before and after fine-tuning in the novel categories (52.0% versus 57.0% ARI).

**(iii)** `AnalogicalNets multi-mem` outperform the single-memory version in many-shot learning with on par few-shot performance.

**(iv)** `Re-DETR3D` adapt few-shot better than `DETR3D`. Still, before weight fine-tuning in the few-shot test set, this memory-augmented variant exhibits worse performance than our single-memory model (46.8% versus 52.0% ARI).

**(v)** A retriever that better recognizes object categories would provide a performance boost, especially in the many-shot setting.

### 4.1.2 EMERGENT CROSS-SCENE CORRESPONDENCE

Within-scene pre-training promotes correspondences between memory and input parts. We show that memory parts can propagate any labels they are associated with to the cross-scene corresponding input parts they decode. We quantify the ability to propagate semantic instance parts in Table 2. All variants of Analogical Networks are trained without any semantic labelling objectives. For reference, we also train the baselines `DETR3D` and `PrototypicalNets` for both object part segmentation

| Method | Novel Categories 1-shot | | Novel Categories 5-shot | | Base Categories Many-shot | |
|---|---|---|---|---|---|---|
| | mIoU | mAP | mIoU | mAP | mIoU | mAP |
| DETR3D* | 21.5 | 18.3 | 30.6 | 27.5 | 55.9 | 53.6 |
| AnalogicalNets single-mem w/o pretrain | 5.0 | 3.3 | 4.7 | 3.9 | 7.8 | 6.2 |
| AnalogicalNets single-mem | 20.4 | 18.2 | 26.0 | 25.0 | 44.3 | 42.0 |
| AnalogicalNets multi-mem | - | - | 27.8 | 25.8 | 49.2 | 47.9 |
| PrototypicalNets* | 27.5 | - | 29.0 | - | 30.0 | - |
| AnalogicalNets OrclCategoryRetrv | 26.2 | 25.2 | 30.2 | 30.2 | 50.6 | 48.7 |

Table 2: **Part Semantic and Part Instance Segmentation performance** on the PartNet benchmark. * indicates training with semantic labels.

| Method | Fine-tuned? | Novel Categories 5-shot ARI (↑) | Chair ARI (↑) |
|---|---|---|---|
| DETR3D trained on "Chair" | ✗ | 28.5 | 76.0 |
| | ✓ | $47.7 \pm 2.31$ | - |
| DETR3D | ✗ | 30.4 | 75.7 |
| | ✓ | $52.7 \pm 1.44$ | - |
| Analogical Networks trained on "Chair" | ✗ | $33.8 \pm 0.89$ | **76.5** |
| | ✓ | $50.4 \pm 1.60$ | - |
| Analogical Networks | ✗ | $\mathbf{52.0 \pm 1.11}$ | 76.3 |
| | ✓ | $\mathbf{57.0 \pm 1.33}$ | - |

Table 3: **Few-shot learning for single-category and multi-category trained models.** Analogical Networks learn better few-shot when trained across all categories, while DETR3D does not improve its few shot learning performance when trained across more categories.

and part labelling. Analogical Networks predict semantic part labels only via propagation from memory part queries $f^m$ that decode parts in the scene, and does not produce any semantic labels for parts decoded by parametric queries $f^p$, so by default it will make a mistake each time a parametric query is used. We found more than 80% of parts are decoded by memory part queries on average, while for AnalogicalNets multi-mem this ratio is 98%. For the few-shot settings in Table 2, all models are fine-tuned on the few given examples. Similar to our model, PrototypicalNets propagate semantic labels of the prototypical part features. We evaluate PrototypicalNets only for semantic segmentation since it cannot easily produce instance segments: if multiple part instances share the same semantic label, this model assumes they belong to the same semantic prototype. Our conclusions are as follows:

**(i)** Analogical Networks show very competitive semantic and instance segmentation accuracy via label propagation through memory part queries, despite having never seen semantic labels at training time. This shows **our model learns cross-scene part associations without any semantic information.**

**(ii)** AnalogicalNets single-mem w/o pretrain has much worse semantic segmentation performance which suggests **within-scene pre-training much helps cross-scene associations to emerge without semantic information.**

**(iii)** PrototypicalNets achieves high 1-shot performance but does not scale with more data and is unable to handle both few-shot and many-shot settings efficiently.

**(iv)** A retriever that better recognizes object categories boosts performance of Analogical Networks over DETR3D in the few-shot settings.

### 4.1.3 MULTI-CATEGORY TRAINING HELPS FEW-SHOT ADAPTATION IN ANALOGICAL NETWORKS

We compare our model and baselines on their ability to improve few-shot learning performance with more diverse training data in Table 3. We train each model under two setups: i) training only on instances of "Chair", which is the most common category—approximately 40% of our training examples fall into this category—(refer to Table 5 for statistics of the training data distribution)

and, ii) training on "all" categories. We test the performance of each model on 5-shot learning. Our conclusions are as follows: **(i) Models trained on a single category usually fail to few-shot generalize to other classes** with or w/o fine-tuning, despite their strong performance on the training class. **(ii)** DETR3D does not improve in its ability to adapt few-shot with more diverse training data, in contrast to Analogical Networks.

In the Appendix, we show the effect of different retrieval mechanisms (6.2, 6.3) and qualitative results on more benchmarks (6.4).

### 4.2    LIMITATIONS

A set of future directions that are necessary for Analogical Networks to scale beyond segmentation of single-object scenes are the following: **(i)** The retriever in Analogical Networks operates currently over whole object memories and is not end-to-end differentiable with respect to the downstream task. Sub-object part-centric memory representations would permit fine-grained retrieval of visual memory scenes. We further plan to explore alternative supervision for the retriever module inspired by works in the language domain (Izacard et al., 2022; Izacard & Grave, 2021). **(ii)** Scaling Analogical Networks to segmentation of complete, multi-object 3D scenes in realistic home environments requires scaling up the size of memory collection. It would further necessitate bootstrapping fine-grained object part annotations, missing from 3D scene datasets (Dai et al., 2017), by transferring knowledge of object part compositions from PartNet. Such semi-supervised fine-grained scene parsing is an exciting avenue of future work. **(iii)** So far we have assumed the input to Analogical Networks to be a complete 3D point cloud. However, this is hardly ever the case in reality. Humans and machines need to make sense of single-view, incomplete and noisy observations. Extending Analogical Networks with generative heads that not only detect analogous parts in the input, but also inpaint or complete missing parts, is a direct avenue for future work.

## 5    CONCLUSION

We presented Analogical Networks, a semi-parametric model for associative 3D visual parsing that puts forward an analogical paradigm of corresponding input scenes to compositions and modifications of memory scenes and their labelled parts, instead of mapping scenes to segments directly. By casting visual parsing as analogical correspondence, Analogical Networks can few-shot learn better than parametric alone models. One-shot, few-shot or many-shot learning are treated uniformly in Analogical Networks, by conditioning to the appropriate set of memories, whether taken from a single, few or many memory exemplars, and inferring analogous parses. We showed Analogical Networks outperform SOTA parametric and meta-learning baselines in few-shot 3D parsing. We further showed correspondences emerge across scenes without semantic supervision, as a by-product of the analogical inductive bias and our within-scene pre-training. In his seminal work "the proactive brain" (Bar, 2007), Moshe Bar argues for the importance of analogies and associations in human reasoning—highlighting how associations of novel inputs to analogous representations in memory can drive perceptual inference. Analogical Networks operationalize these insights in a retrieval-augmented in-context 3D parsing framework, with analogy formation between retrieved memory graphs and input scene segmentations.

## REPRODUCIBILITY STATEMENT

To ensure the reproducibility of the empirical results, we include a pseudo-code of the main model components and training pipeline in the Appendix. We have made our code publicly available on GitHub.

### ACKNOWLEDGMENTS

This work is supported by Sony AI, a DARPA Young Investigator Award, an NSF CAREER award, an AFOSR Young Investigator Award, a Verisk AI Faculty Research Award, and DARPA Machine Common Sense.

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

## 6 APPENDIX

Our Appendix is organized as follows: In Section 6.1, we provide implementation details and pseudo-code for training Analogical Networks. In Section 6.2 we ablate Analogical Networks' performance under varying memory retrieval schemes in 5-shot setting and show qualitative results for the retriever in Section 6.3. We show more results on noisy point clouds (ScanObjectNN (Uy et al., 2019) dataset) in 6.4. We provide extensive qualitative visual object parsing results for single and multi-memory variants of Analogical Networks in section 6.5. Lastly, we discuss additional related literature in Section 6.6.

### 6.1 IMPLEMENTATION DETAILS AND TRAINING PSEUDO CODE

The modulator encodes the input scene $S$ and each retrieved memory scene $M$ into a set of 3D point features using PointNet++ backbone. We encode positional information using rotary 3D positional encodings (Su et al., 2021; Li & Harada, 2021), which have the property that $P(x)^T P(y) = P(y-x)$, where $P$ the positional encoding function and $x, y$ two 3D points. These embeddings are multiplied with queries and keys in the attention operations, thus making attention translation-invariant. For memory queries, we use the centroid of the corresponding memory part to compute positional encodings. For scene-agnostic queries, we use the center of the input object.

Each labelled part $p$ in each memory $M$ is encoded into an 1D feature vector $f_p^M$ by average pooling its point features. Next, the queries (memory and scene-agnostic combined) self-attend and cross-attend to the input point features and update themselves; point features self-attend and cross-attend to queries to also update themselves. This input feature update is a difference from existing detection transformers, where only the queries are updated. We consider 6 layers of self and cross-attention.

Lastly, we upsample the point features to the original resolution using convolutional layers (Qi et al., 2017) and compute an inner product between each query and point feature to compute the segmentation mask for each query. The output of the modulator is a set of $N_q$ segmentation mask proposals and corresponding confidence scores, where $N_q$ is the total number of queries. At training time, these proposals are matched to ground-truth instance binary masks using the Hungarian matching algorithm (Carion et al., 2020). For the proposals that are matched to a ground-truth instance, we compute the segmentation loss, which is a per-point binary cross-entropy loss (Vu et al., 2022; Cheng et al., 2021b). We also supervise the confidence score of each query, similar to (Carion et al., 2020). The target labels are 1 for the proposals matched with a ground-truth part and 0 for non-matched. We found it beneficial to apply these losses after every cross-attention layer in the modulator. At test time, we multiply the per point mask occupancy probability with tiled confidence scores to get a $N_p \times N_q$ tensor ($N_p$ is the number of points); then each point is assigned to the highest scoring query by computing inner product and taking per-point argmax over the queries. The modulator's weights are trained with within-scene and cross-scene training where the modulating memories are sampled from the top-k retrieved memories. During cross-scene training, we further co-train with within-scene training data.

For both stages of training (i.e. within-scene correspondence pre-training and cross-scene training), we use AdamW optimizer (Loshchilov & Hutter, 2017) with an initial learning rate of $2e-4$ and batch size of 16. We train the model for 100 epochs within-scene and 60 cross-scene. For few-shot fine-tuning/evaluation, we use AdamW optimizer with an initial learning rate of $3e-5$ and batch size of 8. We fine-tune for 90 epochs and we report the performance across 10 different episodes, where each episode has a different set of K support samples. We describe Analogical Networks' training details in pseudo-code for within (Algorithm 1) and cross-scene training (Algorithm 2) respectively.

For our `DETR3D` baseline we use same hyperparameters and train for 250 epochs. Training takes approximately 15 and 20 minutes per epoch on a single NVIDIA A100 gpu for `DETR3D` and Analogical Networks respectively. For the multi-memory model we reduce the batch size to 8 and the learning rate to $1e-4$. Each epoch takes around 50 minutes.

### 6.2 PERFORMANCE UNDER VARYING RETRIEVAL SCHEMES

In the few-shot setting, we evaluate the performance of Analogical Networks under varying memory retrieval schemes in Table 4. We compare against a hypothetical *oracle* retriever that can fetch

---

**Algorithm 1:** Pseudo code for within-scene correspondence pre-training of Analogical Networks

---

```
# S: input point cloud, M: memory point cloud, Np: numbers of points in S or M, N:
      sub-sampled points, C: number of feature channels, P: number of parts in M
# augment: a sequence of standard 3D point cloud augmentations
# pc_encoder: point cloud encoder
# Xp(M): ground-truth label assignment of points in parts, copied directly from
      the memory
# part_encoder: Computes the part features using mean pooling
# pos_encode: Adds positional encoding
# upsampler: Upsamples point cloud
# Segmentation_Loss: Cross entropy loss to assign each point to the Hungarian
      matched query.

for S in dataloader: # load a batch with B samples
   M = S # the memory is the un-augmented version
   S = augment(S) # the input is augmented
   # S : B x Np x 3 and M: B x Np x 3
   # Compute point features
   F^S = pc_encoder(S) # B x N x C
   F^M = pc_encoder(M) # B x N x C

   # Initialize memory part queries
   f^M = part_encoder(F^M) # B x P x C

   # Compute positional embedding
   x_pos = pos_encode(F^S)
   y_pos = pos_encode(f^M) # B x P x C

   Loss = 0
   # Do multiple layers of modulation using Self-Attn and Cross-Attn
   for layer in num_layers:
      x = Cross-Attn(x, y, x_pos, y_pos) # B x N x C
      y = Cross-Attn(y, x, y_pos, x_pos) # B x P x C
      x = Self-Attn(x, x_pos) # B x N x C
      y = Self-Attn(y, y_pos) # B x P x C

      X = upsampler(x) # B x Np x C
      point_query_similarity = matmul(normalize(X), normalize(y.T)) # B x Np x P

      Loss += Segmentation_Loss(argmax(point_query_similarity, -1), Xp(M))

   # optimizer step
   loss.backward()
   optimizer.step()
```

---

the most helpful (in terms of resulting ARI) memory for each input. All examined retrievers are category-constrained, i.e., they have access to the object category of the input and retrieve an object of the same category. Our conclusions are as follows: **(i) Analogical Networks with an oracle memory retriever perform better than Analogical Networks using memories retrieved by the retriever**. This suggests that better training of our retriever or exploring its co-training with the rest of our model could have significant impact in improving its performance. **(ii) Considering any of the 5-shot exemplars randomly does worse than using memories retrieved by our retriever.**

## 6.3 QUALITATIVE PERFORMANCE OF THE RETRIEVER

We show qualitative results of the retriever on multiple classes, both seen (Figure 4) during training and unseen (Figure 5). We observe the following: **(i) The retriever considers fine-grained object similarities and not only class information.** To illustrate this, we include two examples for the "Chair" and "Earphone" classes in Figure 4, as well as the "Bed" and "Refrigerator" classes in Figure 5. Different instances of the same category retrieve very different memories, that share both structural and semantic similarities with the respective input point cloud. **(ii) The retriever generalizes to novel classes, not seen during training**, as shown in Figure 5.

---

**Algorithm 2:** Pseudo code for cross-scene training of Analogical Networks

---

```
# S: input point cloud, M: retrieved memory point cloud, Np: numbers of points in
    S or M, N: sub-sampled points, C: number of feature channels, P: number of
    parts in M, target_classes: semantic classes of ground-truth parts of S
# Q: number of learnable scene-agnostic queries
# pc_encoder: point cloud encoder
# Xp_Hungarian: Hungarian matched label assignment of points in S to queries
# yp_Hungarian: Hungarian matched label assignment of queries to GT parts in S
# matched: indices of queries that have been matched to a ground-truth part
# part_encoder: Computes the part features using mean pooling
# pos_encode: Adds positional encoding
# upsampler: Upsamples point cloud
# Objectness_Loss: Binary cross entropy loss to decide which queries (scene-
    agnostic+learnable) would be responsible for decoding parts
# Segmentation_Loss: Cross entropy loss to assign each point to the hungarian
    matched query.
# Semantic_Loss: Cross entropy loss to map hungarian matched queries to semantic
    classes

for S,M in dataloader: # load a batch with B samples
    # S : B x Np x 3 and M: B x Np x 3
    # Compute point features
    F^S = pc_encoder(S) # B x N x C
    F^M = pc_encoder(M) # B x N x C

    # Initialize memory part queries
    f^M = part_encoder(F^M) # B x P x C

    # Compute positional embedding
    x_pos = pos_encode(F^S)
    y = pos_encode(Concatenate(f^M, scene_agnostic_queries)) # B x (P + Q) x C

    Loss = 0
    # Do multiple layers of modulation using Self-Attn and Cross-Attn
    for layer in num_layers:
        x = Cross-Attn(x, y, x_pos, y_pos) # B x N x C
        y = Cross-Attn(y, x, y_pos, x_pos) # B x P x C
        x = Self-Attn(x) # B x N x C
        y = Self-Attn(y) # B x P x C

        X = upsampler(x) # B x Np x C
        point_query_similarity = matmul(normalize(X), normalize(y.T)) # B x Np x (P +
            Q)

        Loss += Segmentation_Loss(armax(point_query_similarity, -1), Xp_Hungarian) +
            Objectness_Loss(y, yp_Hungarian) + Semantic_Loss(y[matched],
            target_classes)

    # optimizer step
    loss.backward()
    optimizer.step()
```

---

| Method | Fine-tuned? | Modulating Memory | Novel Category: 5-shot ARI ($\uparrow$) |
|---|---|---|---|
| | | Random category-constr. | $48.5 \pm 0.76$ |
| | | Retriever category-constr. | $50.3 \pm 1.12$ |
| `AnalogicalNets single-mem` | ✗ | Oracle | $61.9 \pm 1.22$ |
| | | Random category-constr. | $53.5 \pm 1.28$ |
| | | Retriever category-constr. | $55.8 \pm 1.16$ |
| `AnalogicalNets single-mem` | ✓ | Oracle | $62.3 \pm 0.66$ |

Table 4: **Ablations on ARI segmentation performance under varying retrieval schemes for 5-shot on 4 novel categories**.

## 6.4 Evaluation on ScanObjectNN Dataset (Uy et al., 2019)

We test Analogical Networks on ScanObjectNN (Uy et al., 2019), which contains noisy and incomplete real-world point clouds. We split the training into 11 classes seen during training (bag, bin, box, cabinet, chair, desk, door, pillow, shelf, sink, sofa) and 4 unseen (bed, display, table, toilet). Note

| Base Categories | | | | | | | | | | | | Novel Categories | | | |
|---|---|---|---|---|---|---|---|---|---|---|---|---|---|---|---|
| Chair | Display | Storage Furniture | Bottle | Clock | Door | Ear phone | Faucet | Knife | Lamp | Trash Can | Vase | Table | Bed | Dishwasher | Refrigerator |
| 6323 | 928 | 2269 | 436 | 554 | 225 | 228 | 648 | 327 | 2207 | 321 | 1076 | 8218 | 194 | 181 | 187 |

Table 5: **Number of samples per category in the PartNet dataset (Mo et al., 2019).** Note that each sample has annotations for three levels of segmentation granularity.

that ScanObjectNN is not consistently labelled. For example, as we show in rows 5 and 6 of Figure 12, the legs of a chair may be annotated as a single part or multiple parts in the dataset. Although the PartNet dataset provides the "level" information, there is no such information in ScanObjectNN. Therefore we only qualitatively evaluate our model in Figure 12 for base classes and in Figure 13 for novel classes. We can see that our predictions are always plausible and consistent with the retrieved memory, even if the expected label space is different. We additionally visualize retrieval results in Figure 14.

## 6.5 Qualitative Results for Single-Memory and Multi-Memory Analogical Networks

In this section, we show our model's qualitative results for object parsing. In the Figures 6, 7, 8, 9, 10, 11, 12, 13 we use the following 5-column pattern:

- Unlabelled input point cloud

- Memory used for modulation

- Object parsing generated using only the memory part queries (not the scene-agnostic queries). In this column, regions that are not decoded by a memory part query are colored in black.

- Final predicted segmentation parsing using both memory-initialized queries and scene-agnostic queries. Regions that are colored in black in the third column but colored differently in the fourth column are decoded by scene-agnostic queries.

- Input point cloud's ground truth segmentation at the granularity level of the memory.

We qualitatively show the emergence of part correspondence between retrieved memory (column 2) and the input point cloud parsed using memory queries (column 3). Parts having the same color in columns 2 and 3 demonstrate correspondence, i.e. a part in column 2 decodes the part with the same color in column 3. Analogical Networks promote correspondence of parts on both base (Figure 6 bottom and 7) and novel (Figure 6 top and 8) categories. This correspondence is semantic but also geometric, as can be seen in Figure 10. When multiple memories are available, Analogical Networks mix and match parts of different memories to parse the input. Furthermore, we show parsing results for `AnalogicalNets single-mem w/o pretrain` in Figure 11. We observe that all of memory part query are inactive in the parsing stage. This demonstrates the utility of within-scene pre-training, as without this pre-training part correspondence does not emerge, as shown in Figure 11. Lastly, we show that Analogical Networks generalize to noisy and incomplete point clouds in ScanObjectNN.

## 6.6 Additional related work

**Neural-symbolic models** Analogical Networks are a type of neural-symbolic model that represents knowledge explicitly, in terms of structured visual memories, where each one is a graph of part-entity neural embeddings. A structured visual memory can be considered the neural equivalent of a FRAME introduced in (Minsky, 1985), *"a graph of nodes and their relations for representing a stereotyped situation, like being in a certain kind of living room, or going to a child's birthday party"*, to quote Minsky (1985). FRAME nodes would operate as "slots" to be filled with specific entities, or symbols, in the visual scene. Symbol detection would be carried out by a separate state estimation process such as general-purpose object detectors (Zhou et al., 2022), employed also by recent neuro-symbolic models (Yi et al., 2018; Mao et al., 2019; Yi et al., 2019). In-the-wild detection of symbols (e.g., chair handles, faucet tips, fridge doors) typically fails, which is the reason why these earlier symbolic systems of knowledge, largely disconnected from the sensory input, have not been widely adopted. Analogical Networks take a step towards addressing these shortcomings by

including symbol detection as part of inference itself, through a top-down modulation that uses the context represented in the memory graph, to jointly search for multiple entities and localize them in context of one another.

**3D instance segmentation** has been traditionally approached as a clustering problem (Chen et al., 2009; Sidi et al., 2011). Point-based methods learn either translation vectors mapping every point to its instance's center (Jiang et al., 2020; Chen et al., 2021; Vu et al., 2022) or similarities across points (Wang et al., 2018a; Zhang & Wonka, 2021), followed by one or more stages of clustering. Similarly, (Wang et al., 2021; Jones et al., 2022) oversegment the point cloud into small regions and then merge them into parts. Yu et al. (2019) recursively decompose a point cloud into segments of finer resolution. (Mo et al., 2019; Sun et al., 2022) learn representative vectors that form clusters by voting for each point. However, these approaches usually assume a fixed label space and need to train a separate model for each sub-task. In contrast, we employ Detection Transformers (Carion et al., 2020) for instance segmentation by repurposing the query vectors to act as representative vectors. We extend this set of queries with memory-initialized queries, enabling in-context reasoning. This allows us to train one model across all categories. As our results show, in absence of such memory contextualization, training one model across multiple categories hurts generalization (Table 3).

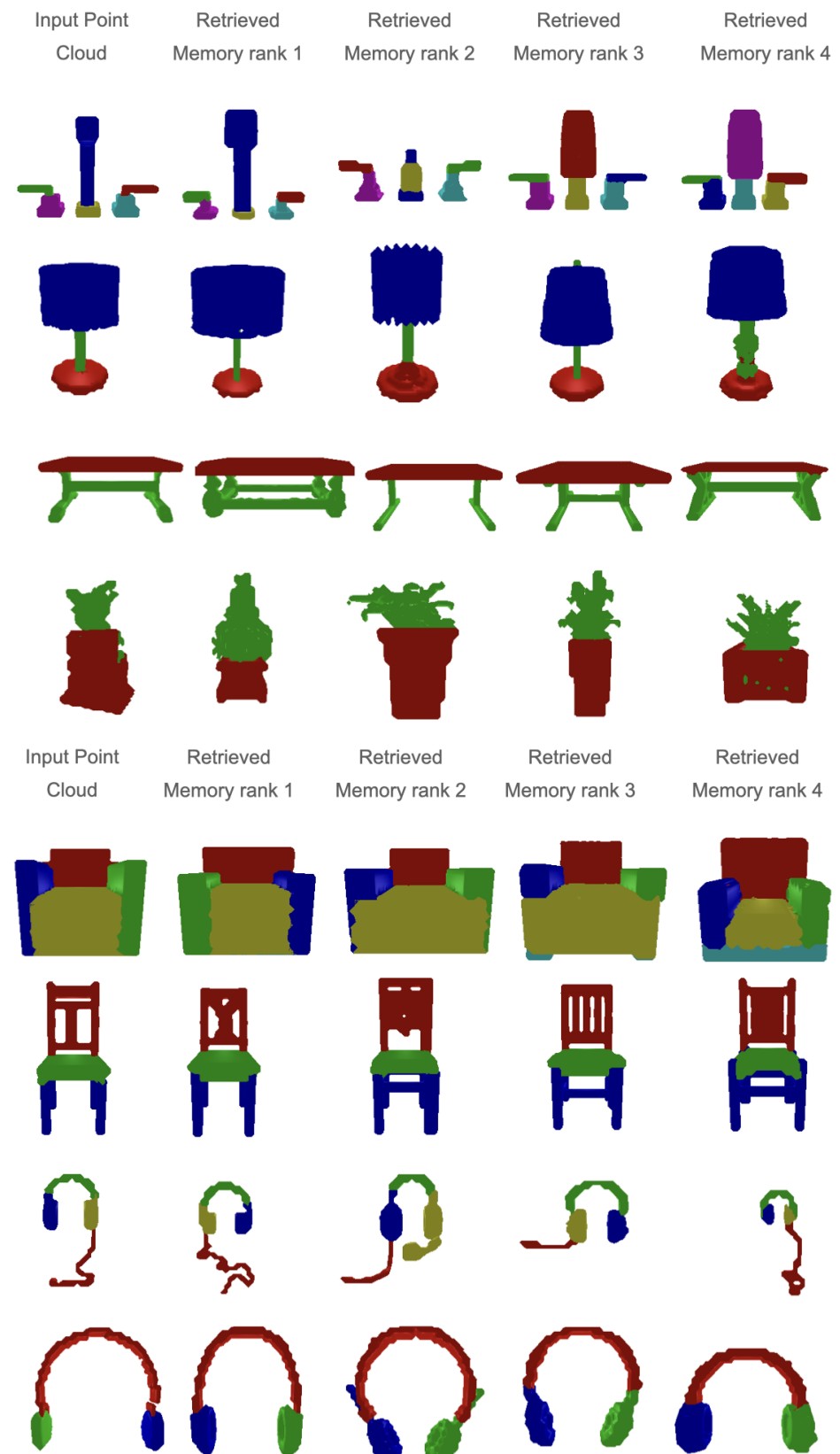

Figure 4: Top-4 retrieved results for each input point cloud. Examples from base classes of PartNet (Mo et al., 2019) dataset. Note that instances of the same category can retrieve different memories, focusing on structural similarity and not only semantic.

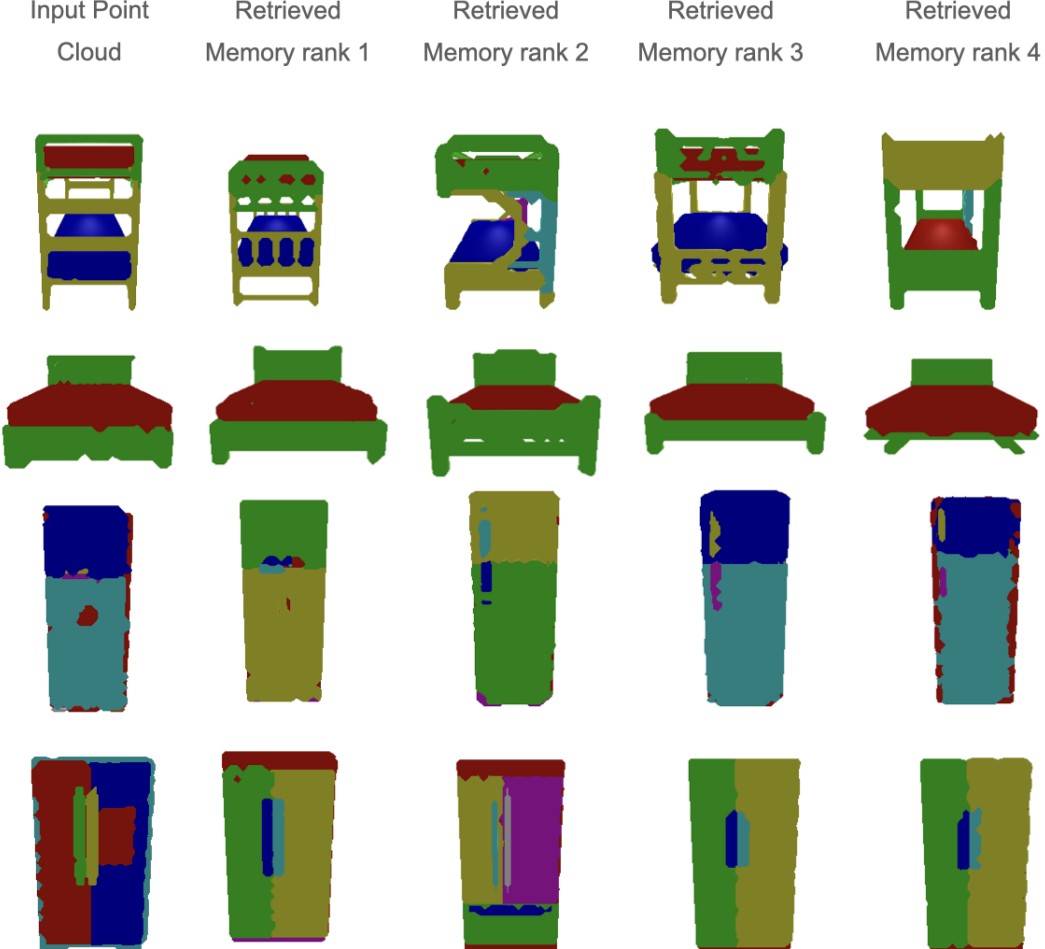

Figure 5: Top-4 retrieved results for each input point cloud. Examples from novel classes of PartNet (Mo et al., 2019) dataset. Note that instances of the same category can retrieve different memories, focusing on structural similarity and not only semantic. This behavior generalizes to novel classes as well, even if the model has never seen such geometries before.

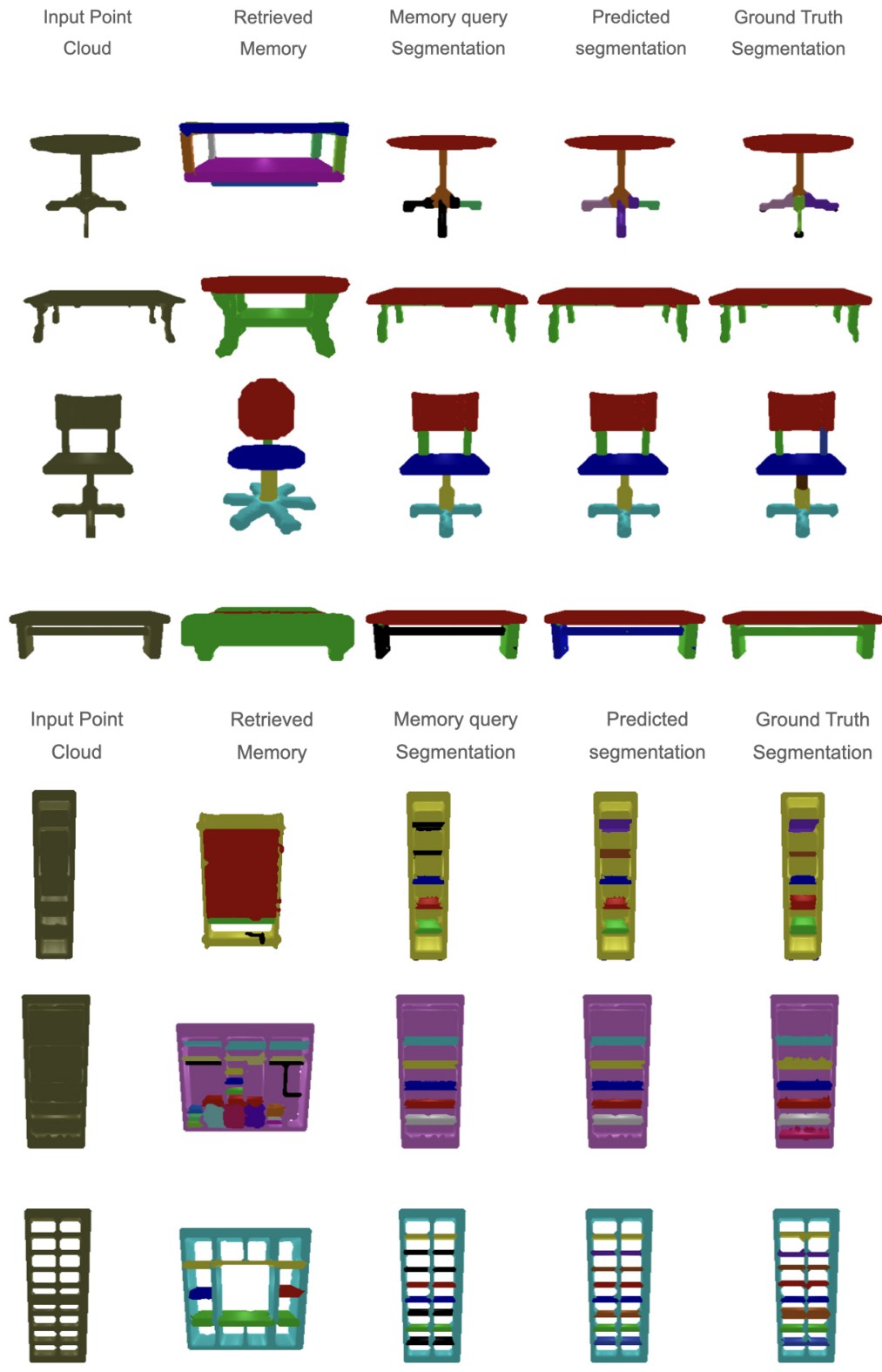

Figure 6: More qualitative object parsing results using Analogical Networks.

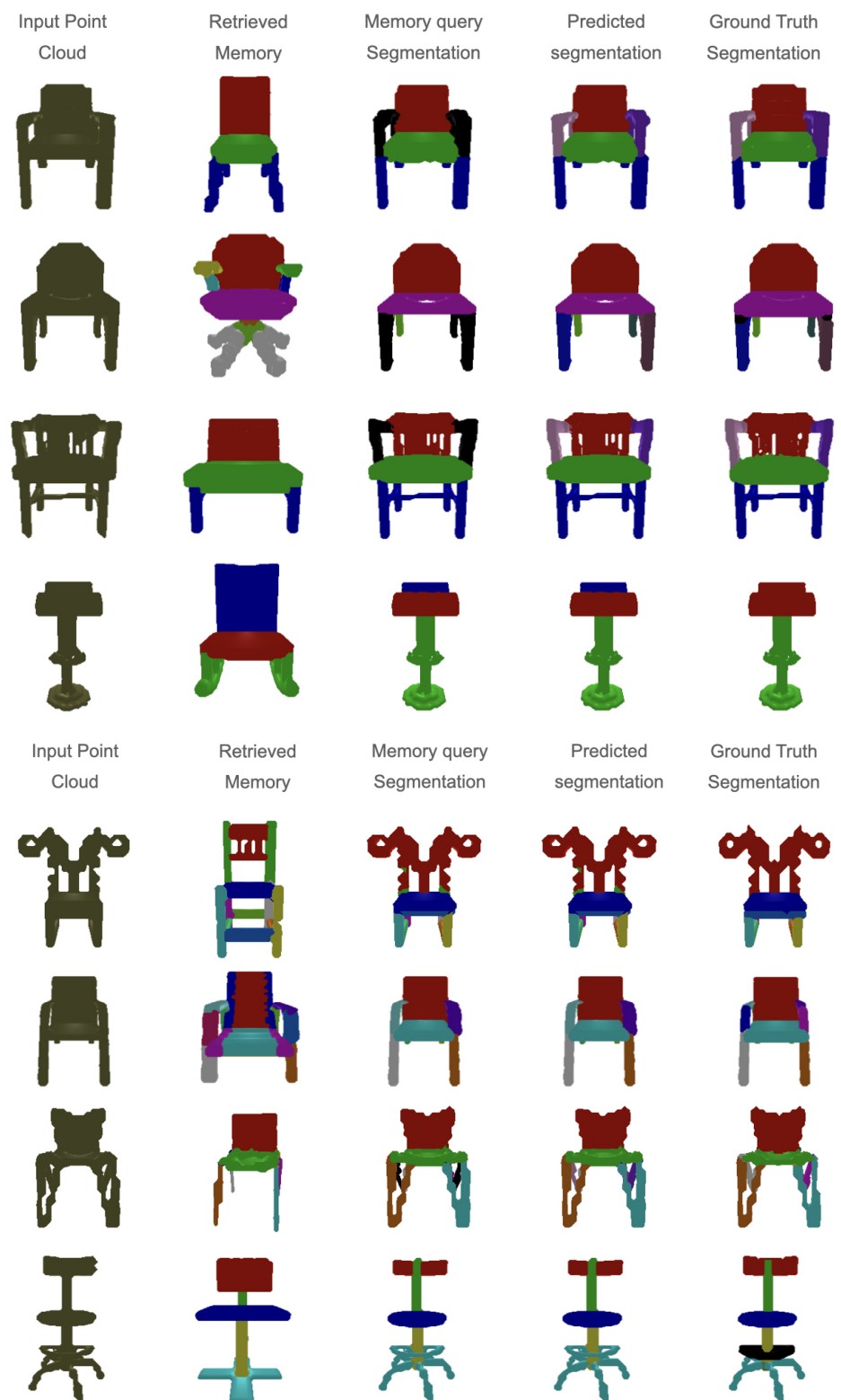

Figure 7: More qualitative object parsing results that are predicted by Analogical Networks.

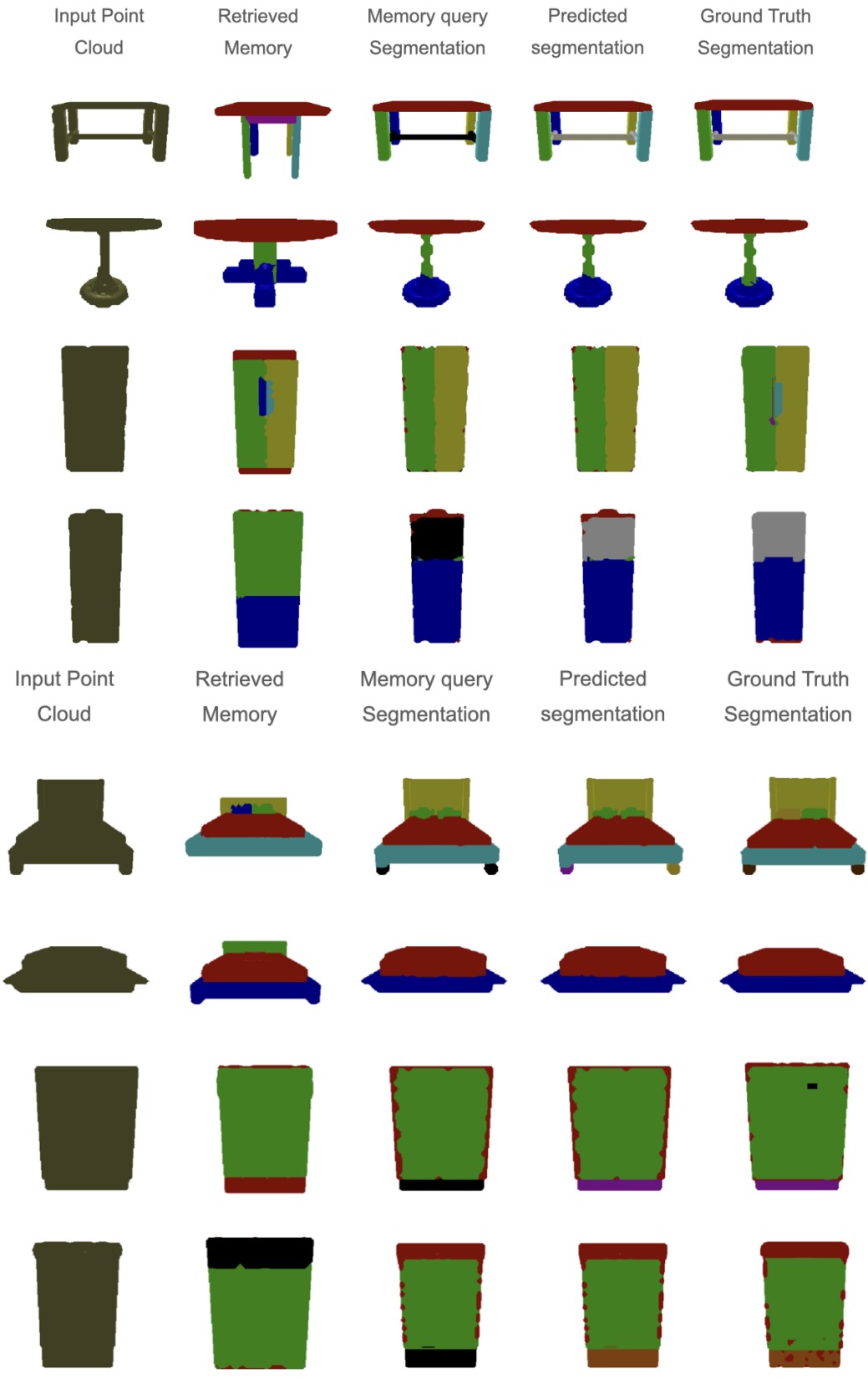

Figure 8: Qualitative results on novel category samples from PartNet dataset (Mo et al., 2019) using Analogical Networks **without fine-tuning**.

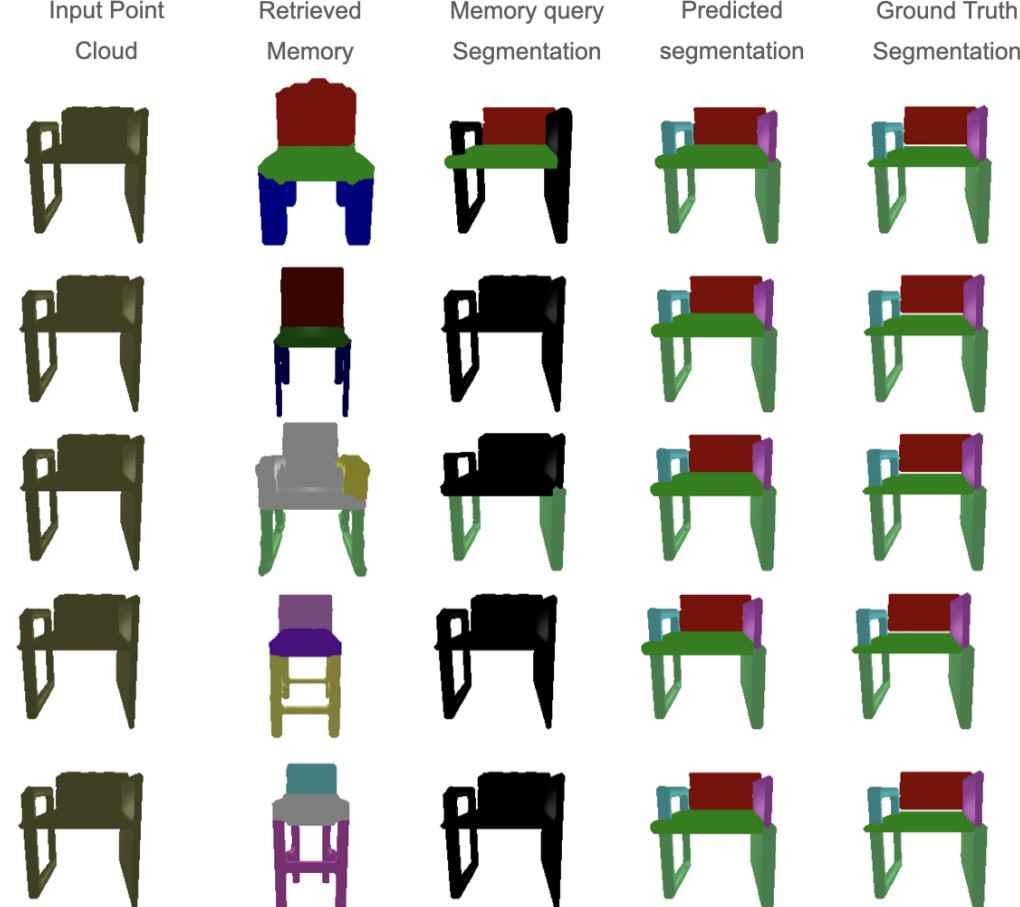

Figure 9: Modulation using multi-memory Analogical Networks. Our model takes as input 5 different memories simultaneously and then parses the object. Each row shows the effect of a different memory. All memories decode simultaneously and we show which part each one decodes in the third column. In the fourth column we show the combined predictions of all memories and scene-agnostic queries.

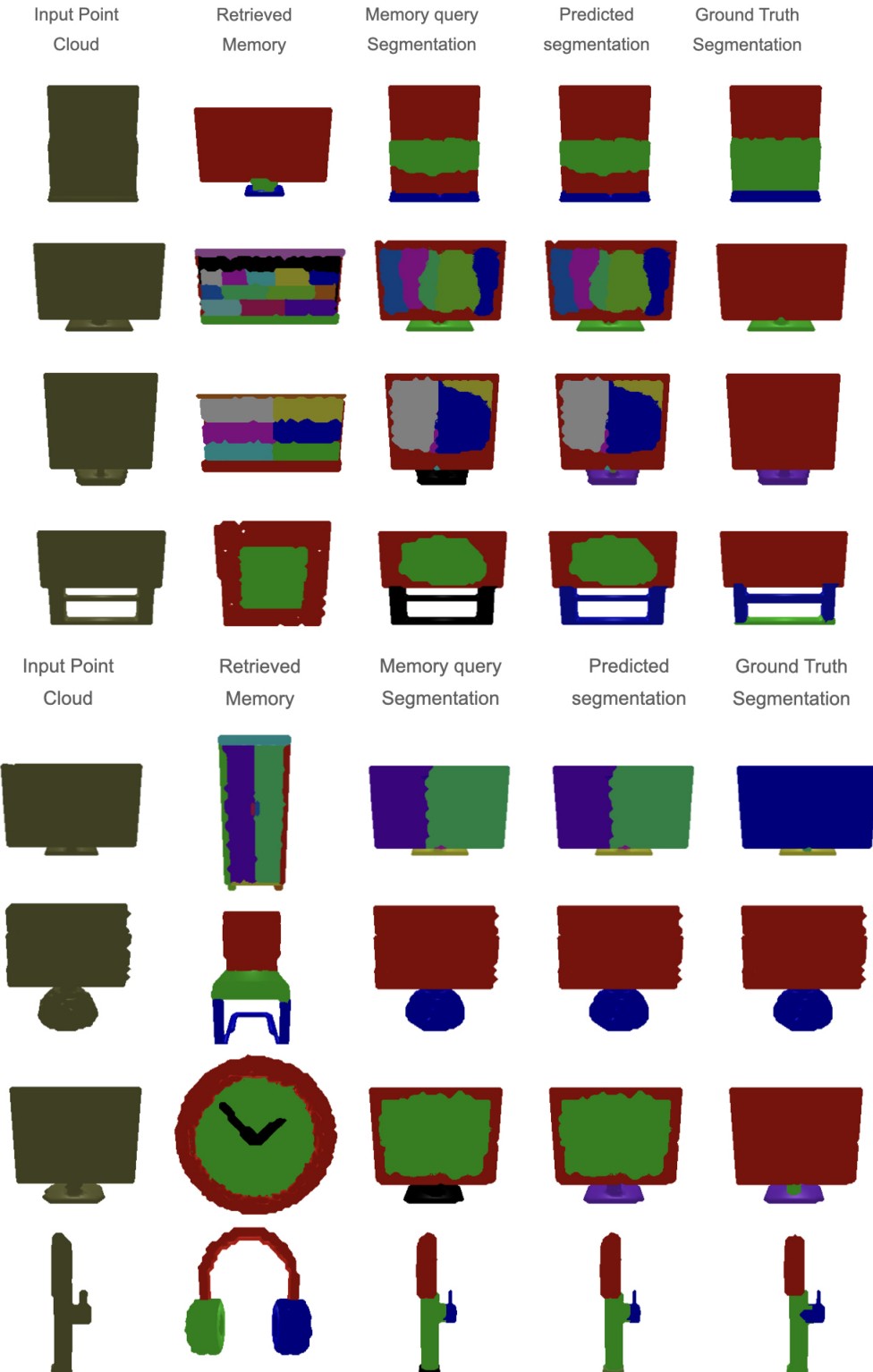

Figure 10: To qualitatively evaluate the effect of modulation in parsing, we modulate the input point cloud with a different category object and show its corresponding object parsing that is predicted by Analogical Networks. The model is able to generalize geometric correspondences across instances of totally different classes, e.g. display and clock.

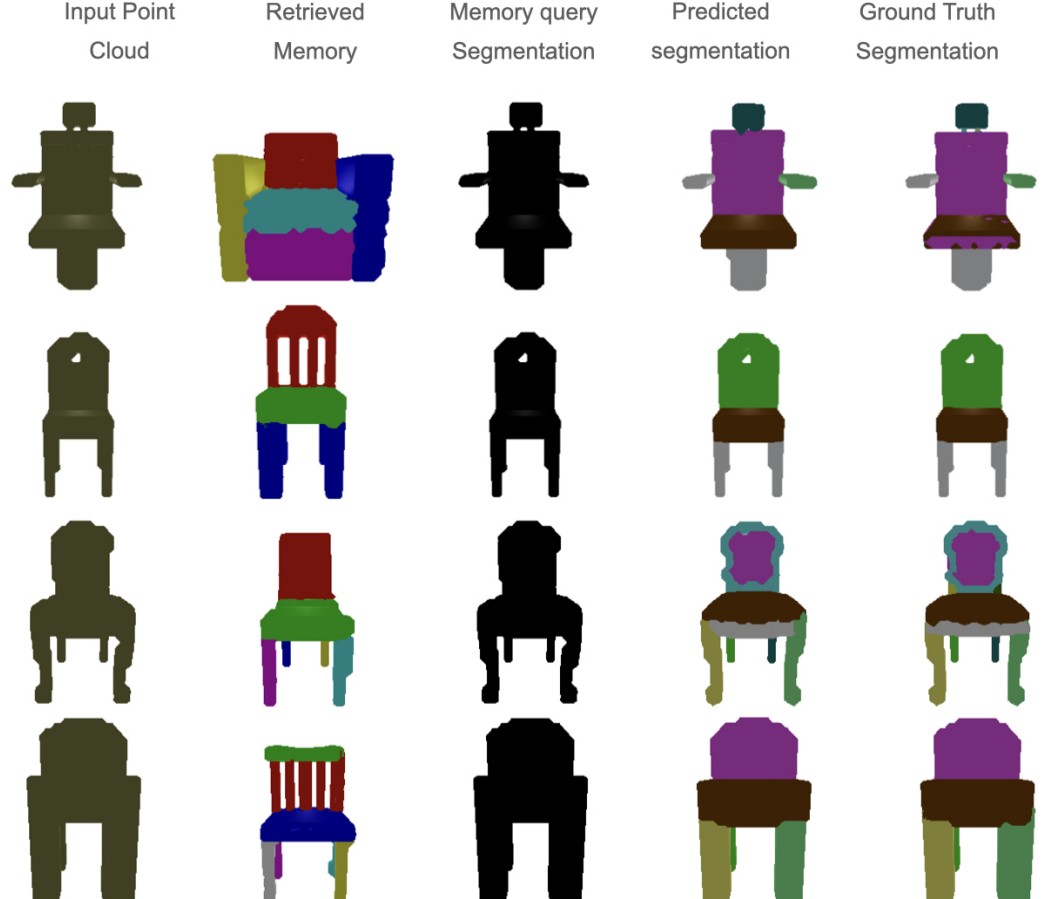

Figure 11: We show the parsing of input point cloud using `AnalogicalNets single-mem w/o pretrain`. Most regions are black in column 3, denoting that memory part queries do not decode anything and everything is being decoded by scene-agnostic queries. This highlights the role of within-scene pre-training for the emergence of part correspondence.

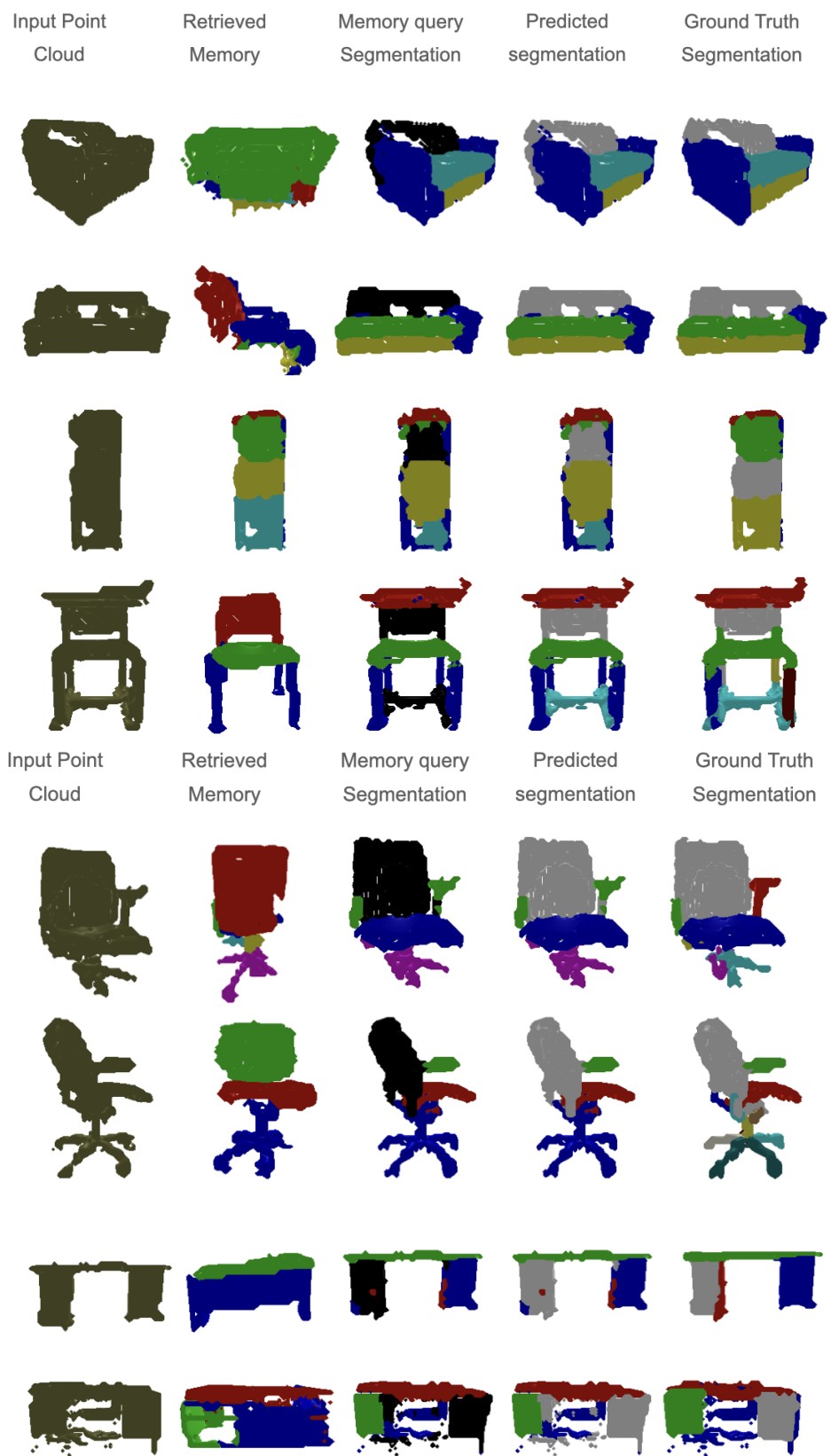

Figure 12: Results on base category samples from ScanObjectNN (Uy et al., 2019) using Analogical Networks.

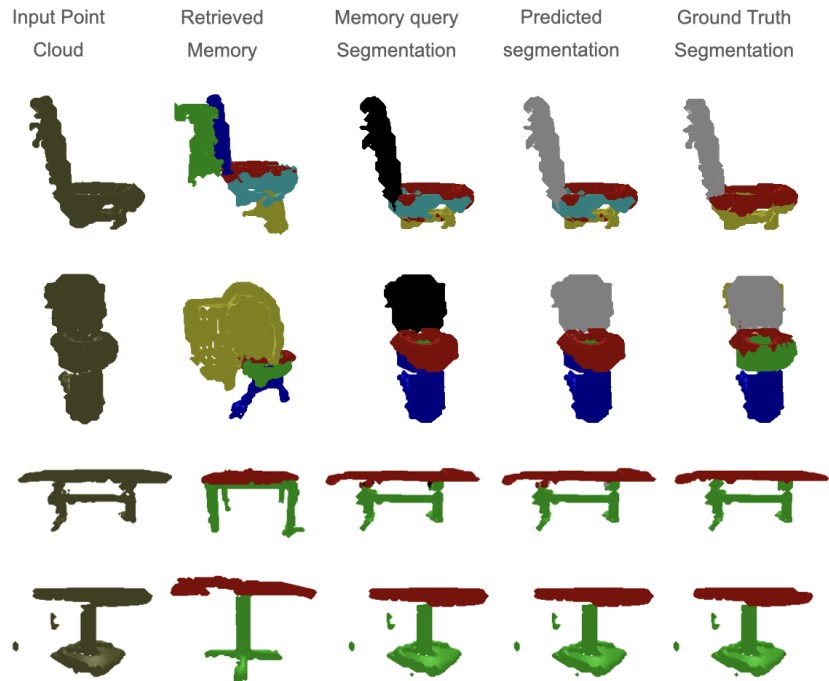

Figure 13: Results on novel category samples from ScanObjectNN (Uy et al., 2019) using Analogical Networks.

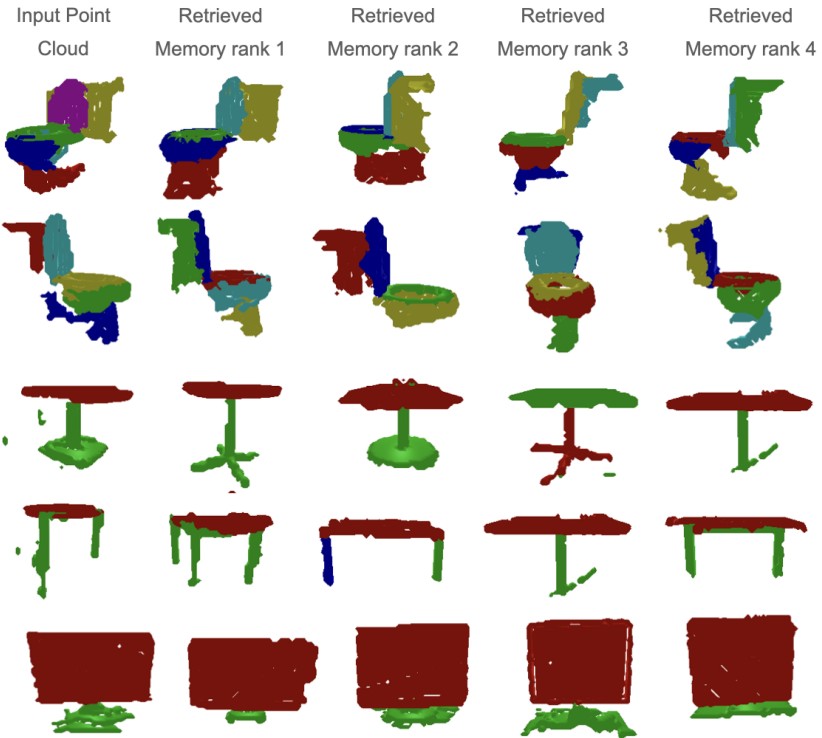

Figure 14: Top-4 retrieved results for the input point cloud from ScanObjectNN (Uy et al., 2019) dataset.

