# OpenReview forum: "Analogy-Forming Transformers for Few-Shot 3D Parsing"
_ICLR.cc/2023/Conference — ICLR 2023 poster_

### Official Review · Reviewer_mbsn · 2022-10-21

**Confidence:** 4
**Correctness:** 3
**Technical Novelty And Significance:** 3
**Empirical Novelty And Significance:** 2
**Recommendation:** 5

**Clarity, Quality, Novelty And Reproducibility:**

The novelty of the whole paper is limited. The clarity is good and quality it not satisfying.  Reproducibility seems good.

**Strength And Weaknesses:**

Strength:
1. All figures and tables are in good illustration
2. The idea is simple and easy to be implemented.

Weakness:
1. The abstract should be revised. The main content is not close to the title, such as 3D parsing is missing in the abstract.
2. Overclaim issues. In the introduction section, the authors claims that  This allows us to train one model across all tasks. However, only one task is evaluated meanwhile this 3d parsing is not a representative few shot learning task.
3. Besides, the idea of Memory-augmented neural networks for few shot learning has been studied in the previous works[1]. From this point, the novelty of the whole method is limited.
[1] Santoro, A., Bartunov, S., Botvinick, M., Wierstra, D., Lillicrap, T.: Meta-learning with memory-augmented neural networks. In: ICML (2016)
4. In experimental section, the setting of many shots are nonsense. This paper focuses on few-shot learning.
5. The experimental results are not sufficient. The compared methods are missing.


**Summary Of The Paper:**

This paper reformulates few shot learning about 3D parsing from  analogy-driven prediction view. The authors design an  Analogical Networks that are comprised of an encoder, retriever and modulator sub-modules. Experiment are performed on benchmark for 3D object segmentation.

**Summary Of The Review:**

This paper presents a new view for few shot learning. However, the idea is incremental from the memory based few shot learning solution. The experimental results and analysis are not sufficient. Overall, the quality of the whole paper can not reach the level of the top AI conference.

---

> ### Author Response · Authors · 2022-11-13
> **Response to Reviewer mbsn (Part 1)**
>
> >Q1: The abstract should be revised. The main content is not close to the title, such as 3D parsing is missing in the abstract.
>
> A1: Following your suggestion, we have **revised the abstract** to be: “Despite recent breakthroughs in the applications of deep neural networks in visual perception, one persistent challenge is few-shot learning. Works in the area of few-shot visual learning mostly address the task of coarse image classification, while fine-grained 3D visual parsing, although useful for scene understanding and interaction with the physical world, remains underexplored. Thus far, a separate neural model is trained to parse each semantic category, which hinders knowledge sharing across objects and few-shot generalization. We present Analogical Networks, a model that casts fine-grained visual parsing into analogical inference: instead of mapping input scenes to part labels, which is hard to adapt in a few-shot manner to novel inputs, our model retrieves related scenes from memory and their corresponding part structures, and predicts analogous part structures in the input scene, via an end-to-end learnable modulation mechanism. By conditioning on more than one memory, compositions of structures are predicted, that mix and match parts from different visual experiences. We show that Analogical Networks excel at few-shot 3D parsing, where instances of novel object categories are successfully parsed simply by expanding the model’s memory, without any weight updates. Analogical Networks outperform existing state-of-the-art detection transformer models for part segmentation, as well as paradigms of meta-learning and few-shot learning. We show that part correspondences emerge across memory and input scenes by simply training for a label-free segmentation objective, as a byproduct of the analogical inductive bias.”
>
> ***
>
>
> >Q2: Overclaim issues. In the introduction section, the authors claims that this allows us to train one model across all tasks. However, only one task is evaluated meanwhile this 3d parsing is not a representative few shot learning task.
>
> A2: We have **edited the paper (Section 1) to refine our claim**: “In this work, a task refers to parsing objects of different categories or segmentation granularity“. Our definition of a task is similar to the definition that few-shot image classification works use. They map images to a new set of classes. We map point clouds to new sets of parts. Our model handles multiple such sets of parts simultaneously, guided by the retrieved memories. Additionally, the dataset we use, PartNet, has 3 different levels of segmentation annotations, from coarse to fine-grained level. We train a single model that adapts to the segmentation level by retrieving memories of different label granularity.
>
> In addition, we respectfully disagree that 3D parsing is not a representative few-shot learning setting, as also shown in [1]. We agree that it is under-explored, but this adds value to our contributions.

---

> > ### Author Response · Authors · 2022-11-13
> > **Response to Reviewer mbsn (Part 2)**
> >
> > >Q3: Besides, the idea of Memory-augmented neural networks for few shot learning has been studied in the previous work of Santoro et al. From this point, the novelty of the whole method is limited.
> >
> > A3: The seminal work of Santoro et al. [2] is an inspiration for our work and we cite it in our related work section. They extended the idea of [3] to use external memory to store and retrieve information in order to adapt to the current input for the visual domain as well.
> >
> > However, **we highlight several fundamental differences between our work and Santoro et al. [2]**.
> > - For [2] and [3], the memory is a fixed set of vectors (a “NxM matrix” to quote [3]) that are updated through write operations. These vectors do not correspond to explicit previous experiences of the model; they are simply useful representations that, when retrieved, help to modulate the processing of the current input. In contrast, we store encodings of visual exemplars in a graph structure, so each memory item is a graph, not a single vector.
> > - We train the memory for correspondence: we retrieve and modulate through cross and self attention operations that compute a correspondence between the memory nodes and the object parts in the input scene. Our memory-initialized queries directly decode analogous parts in the input point cloud. We show correspondences emerge between memory nodes and input parts through useful memory-modulated pretraining. There is no notion of correspondence in memory-augmented neural networks.
> > - Our model can learn fast by memory expansion. While [2] and [3] can in theory extend the memory by increasing the number of elements N, they need to re-train their model and memory bank so that the new entries learn useful representations. In our case though, we can dynamically scale the memory simply by appending new annotated examples of completely novel classes and label spaces, without having to re-train or even query our network to predict read-write weights.
> > - [2] tackles classification of 2D grayscale images, while we address part segmentation of 3D point clouds. To the best of our knowledge we are the first to explore the idea of using external memory to adapt in a few-shot manner for structure prediction tasks on the visual domain, which is a non-trivial extension.
> >
> > ***
> >
> > >Q4: In the experimental section, the setting of many shots are nonsense. This paper focuses on few-shot learning.
> >
> > A4: We believe that having good performance on both setups (many-shot and few-shot) is important, especially from the perspective of a continual learning agent that is trained on a fixed set and then faces new scenarios after deployment. In fact, the world is full of such cases of long tails: some classes have many examples and tons of other classes have few. It is impossible to decide a priori whether a real-world scenario would be many or few-shot, so it makes sense for us to have **a model that can unify these two setups, few and many-shot**, as we state in our contributions. To the best of our knowledge, many other few-shot papers [4-6] show many-shot results as well.

---

> > > ### Author Response · Authors · 2022-11-13
> > > **Response to Reviewer mbsn (Part 3)**
> > >
> > > >Q5: The experimental results are not sufficient. The compared methods are missing.
> > >
> > > A5: We **added Table 4**, in our appendix that evaluates our model on instance segmentation with labels on PartNet. We **outperform all compared methods (15.2% average boost over [7], 22.5% boost over [8])**, despite the fact that they train on each class and level separately and despite the fact that our model has not seen any training examples for the novel classes.
> > >
> > > Additionally, we **trained the PartNet model of [9] on our setup** of multiple classes and segmentation levels (see **revised Table 1**). Same as in 3D-DETR, we provide the model with a level id embedding. We find that its performance is significantly worse than all our variants and other baselines, specifically **15% worse than Analogical Networks in many-shot and 17% in 1-shot**.
> > >
> > > To the best of our knowledge, most part-segmentation approaches cannot adapt few-shot or handle multiple classes and segmentation levels with a single model. To this end, we compared Analogical networks to baselines that have this capability, as we also explained to reviewer MKmk, in A1. Please let us know if you think that we missed some important comparison.
> > >
> > > ***
> > >
> > > Thank you and please let us know if there are any additional questions. We would appreciate your feedback.
> > >
> > > References:
> > >
> > > [1] Zhao et al. 2021, “Few-shot 3D Point Cloud Semantic Segmentation”
> > >
> > > [2] Santoro et al. 2016, “Meta-Learning with Memory-Augmented Neural Networks”
> > >
> > > [3] Graves et al. 2014, “Neural Turing Machines”
> > >
> > > [4] Gidaris et al. 2018, “Dynamic Few-Shot Visual Learning without Forgetting”
> > >
> > > [5] Lifchitz et al. 2019, “Dense Classification and Implanting for Few-Shot Learning”
> > >
> > > [6] Liu et al. 2019, “Large-Scale Long-Tailed Recognition in an Open World”
> > >
> > > [7] Sun et al. 2022, “Semantic segmentation-assisted instance feature fusion for
> > > multi-level 3d part instance segmentation”
> > >
> > > [8] Zhang et al. 2021, “Point Cloud Instance Segmentation using Probabilistic Embeddings”
> > >
> > > [9] Mo et al. 2019, “PartNet: A large-scale benchmark for fine-grained and hierarchical part-level 3D object understanding”

---

> > > > ### Comment · Reviewer_mbsn · 2022-11-20
> > > > **For experimental completeness**
> > > >
> > > > In the revised manuscript, the authors add table 4 for experimental comparison with other methods. However, as a top conference paper, both  Mkmk and I point out that this paper does not compare with other 3D part segmentation methods or clearly place their approach in the context of part segmentation papers.  This is a big weakness for a computer vision paper. As table 4 should be reported in the first round submission. The authors reported this result in the revised manuscript, this is not fair as the authors use more time to revise the manuscript than other papers.
> > > >
> > > > Besides, I have checked the reported results in Table4, the compared methods are not representative [https://paperswithcode.com/dataset/partnet]. Many methods that achieve better performance than the proposed method are missing.  The authors only select some methods with inferior performance. This is not fair and convincing.
> > > >
> > > > Furthermore, as pointed by MKmk, there are lots of point cloud instance segmentation datasets, such as ScanNet. As a top conference paper, it is not sufficient to only adapt one dataset for evaluation.
> > > >
> > > > Take these points into consideration, I maintain my initial score for this submission.

---

> > > > > ### Author Response · Authors · 2022-11-22
> > > > > **Response on experimental completeness**
> > > > >
> > > > > We respectfully disagree with the claim that “many methods that achieve better performance are missing”- to the best of our knowledge, [1] is the current state-of-the-art for **part instance segmentation** on PartNet. We outperform it by an absolute 15.2%, averaged across all tested levels and classes, as shown in Table 4.
> > > > >
> > > > > After thoroughly searching for other baselines, we also found [2], that performs on par with (slightly worse than) [1]. The authors report results on level 3 only and we include a comparison here:
> > > > >
> > > > >
> > > > >
> > > > > | | SAIF [1] | ICM [2] | 3D-DETR | Ours |
> > > > > | :----: | :----: | :----: | :----: | :----: |
> > > > > | Bed | 40.9 | 20.7 | 28.0 | **43.8** |
> > > > > | Bottle | 55.9 | 55.3 | 65.8 | **69.0** |
> > > > > | Chair | 38.2 | 46.9 | 65.0 | **68.6** |
> > > > > | Clock | 37.1 | 29.9 | 31.4 | **52.1** |
> > > > > | Dishwasher | **56.5** | 52.8 | 42.9 | 55.2 |
> > > > > | Display | 87.4 | 86.7 | 83.2 | **92.6** |
> > > > > | Door | 41.3 | 28.9 | 46.1 | **56.3** |
> > > > > | Earphone | 53.7 | 45.7 | **59.2** | **59.2** |
> > > > > | Faucet | 59.1 | 56.3 | 69.6 | **73.2** |
> > > > > | Knife | 48.8 | 52.1 | **71.1** | 62.0 |
> > > > > | Lamp | 21.7 | 51.0 | 55.5 | **55.8** |
> > > > > | Fridge | 44.1 | **49.3** | 32.7 | 43.0 |
> > > > > | StorageF | 44.0 | 55.5 | 55.1 | **68.3** |
> > > > > | Table | 28.9 | 40.0 | 40.0 | **54.2** |
> > > > > | Trash Can | 51.3 | 29.5 | 47.1 | **79.0** |
> > > > > | Vase | 54.6 | 59.8 | 78.8 | **87.7** |
> > > > > | | | | | |
> > > > > | **Average** | 47.7 | 47.5 | 54.4 | **63.7** |
> > > > >
> > > > >
> > > > > We additionally observe in the table above that the baseline we consider (3D-DETR) outperforms [1] and [2] by a significant margin. We would be happy to compare with other suggested baselines.
> > > > >
> > > > > Lastly, we kindly refer the reviewer to our revised appendix (Section 6.7), where we added experiments on the ScanObjectNN dataset [3] that is derived from ScanNet [4] and SceneNN [5]. We outperform the baselines, by 1.2% and 17.5% in many-shot, and by 2.2% and 22.4% in few-shot without fine-tuning.
> > > > >
> > > > >
> > > > > [1] Sun et al. 2022, "Semantic segmentation-assisted instance feature fusion for multi-level 3d part instance segmentation"
> > > > >
> > > > > [2] Chu et al. 2022, "ICM-3D: Instantiated Category Modeling for 3D Instance Segmentation"
> > > > >
> > > > > [3] Uy et al. 2019, "Revisiting Point Cloud Classification: A New Benchmark Dataset and Classification Model on Real-World Data"
> > > > >
> > > > > [4] Dai et al. 2017, "ScanNet: Richly-annotated 3D Reconstructions of Indoor Scenes"
> > > > >
> > > > > [5] Hua et al. 2016, "SceneNN: A Scene Meshes Dataset with aNNotations"

---

> > > ### Comment · Reviewer_mbsn · 2022-11-20
> > > **About A3**
> > >
> > > The memory difference between the proposed method and the mentioned previous works [2,3] is not convincing. There are no experimental or theoretical discussions about the advantages of the proposed memory.

---

> > > > ### Author Response · Authors · 2022-11-22
> > > > **Response about A3**
> > > >
> > > > As discussed in the response, Analogical Networks can adapt to novel classes in a few-shot manner simply by expanding their memory and using it for in-context prediction, **without retraining**. In contrast, Memory-augmented neural networks [2,3] need to retrain their memory embeddings in order to adapt to novel classes, which would require more samples, and hence they are unsuitable for few-shot learning. In addition, for our method **part correspondence emerges** between retrieved memory parts and input parts. This allows us to **propagate the labels** of the memory parts to the input parts without having to train with different semantic label spaces.
> > > >
> > > > Our paper considers two baselines that have memory, Prototypical Networks and Analogical Networks w/o memory part queries. Prototypical networks may be closer to the use of memory in [2], but instead of memorizing whole image embeddings, this baseline memorizes part embeddings, since we tackle the task of 3D object segmentation and [2] tackles the task of coarse image classification. Analogical Networks considerably outperform both baselines and employ different ways of training and using the memory during inference than previous works.
> > > >
> > > > [2] Santoro et al. 2016, “Meta-Learning with Memory-Augmented Neural Networks”
> > > >
> > > > [3] Graves et al. 2014, “Neural Turing Machines”

---

### Official Review · Reviewer_feEi · 2022-10-24

**Confidence:** 3
**Correctness:** 4
**Technical Novelty And Significance:** 3
**Empirical Novelty And Significance:** 3
**Recommendation:** 8

**Clarity, Quality, Novelty And Reproducibility:**

Strength:
  + Good presentation in general on the motivation, the proposed method, and the experiments.

Weaknesses:
 + The description of the pretraining phase and the regular training phase is lacking clarity and details. As mentioned before, it would be difficult for audience to reimplement this work just following the text.
 + It could be difficult for audience to reimplement this work, as it is comprised of many components and requires many phases of training. Therefore to really facilitate the adoption of this framework, I strongly recommend the author to make their code for training and evaluation both available to the public, as promised in the paper as well.

**Details Of Ethics Concerns:**

No ethics concern from me.

**Strength And Weaknesses:**

Strength:
  + The proposed Analogical Networks look novel to me. It opens new directions to pursue and poses interesting questions to answer.
  + The performance gain in few-shot learning settings is significant.

Weaknesses:
 + Although the author has documented some limitations in their work already, it is worth reiterating here (1) the retriever is not end-to-end differentiable (2) the retriever is not scalable due to operating over the whole object memories.

**Summary Of The Paper:**

This work devised a novel framework called *Analogical Networks* for *few-shot* 3D object part parsing. The framework works as follows: (1) given an input point cloud (of novel object type), the top k closest labeled object instances are retrieved from memory set (2) these objects and their labels are used as context information to modulate the 3D parsing of the novel object. Experiments show this framework can work better than previous meta-learning approaches in *few-shot* learning settings. Many ablation studies are also provided for better understanding of the strength and the limitation of this proposed framework.

**Summary Of The Review:**

I agree with the author on their motivation and the proposed approach. This framework looks novel to me and presents many interesting future directions. Therefore I would recommend for accepting this paper, and hope the authors release their code as promised to the facilitate the adoption and the adaption of this framework.

---

> ### Author Response · Authors · 2022-11-13
> **Response to Reviewer feEi**
>
> Thank you for your review and your appreciation of our contributions. We attempt to address your remaining concerns below. We have updated our draft and highlighted the major edits in the main paper with orange text.
>
>
> >Q1: Although the author has documented some limitations in their work already, it is worth reiterating here (1) the retriever is not end-to-end differentiable (2) the retriever is not scalable due to operating over the whole object memories.
>
> A1: We acknowledge these limitations and, as we said in the paper, we believe that these are important avenues for future work.
> Specifically, we are exploring ways to make the retriever differentiable by combining within-batch memories with losses that are differentiable with respect to the retriever’s features, following [1], a language paper that uses very large (hundreds of millions) external question-answer memories.
> We are further exploring sub-object, part-centric, memory representations and the application of the analogical framework to whole scenes.
>
> ***
>
> >Q2: The description of the pretraining phase and the regular training phase is lacking clarity and details.
>
> A2: We **edited the last subsection of Section 3** in the main paper to better explain the training procedure: “fter within-instance pre-training (Algorithm 1), we maintain two copies of the encoder weights: one copy acts as the encoder of the retriever and is frozen. The other copy is used as the encoder for modulation and is further trained cross-instance, i.e. with the modulating memory being sampled from the top-k retrieved memories.”.
>
> Additionally, we **edited figures 2 and 3** to improve clarity on the role of the encoders and the retriever.
>
> For clarity, we have **added the pseudo-code** in the appendix **to describe the within-instance training** separately.
>
> ***
>
> >Q3: It could be difficult for the audience to reimplement this work. I strongly recommend the author to make their code for training and evaluation both available to the public, as promised in the paper as well.
>
> A3: Yes, we agree and we will upload a clean version of our code along with our trained checkpoints upon publication.
>
> ***
>
> Thank you again. We agree on the importance of reproducibility. Please let us know if there are any additional questions.
>
> References:
>
> [1] Izacard et al. 2022, “Few-shot Learning with Retrieval Augmented Language Models”

---

### Official Review · Reviewer_MKmk · 2022-10-28

**Confidence:** 4
**Correctness:** 3
**Technical Novelty And Significance:** 3
**Empirical Novelty And Significance:** 3
**Recommendation:** 8

**Clarity, Quality, Novelty And Reproducibility:**

Clarity:
1. For the emergent part-correspondences experiment it appears that 3D-DETR is comparable in performance to Analogical Networks at 1-shot and 5-shot, but significantly lower at many-shot. The trend is the opposite for the part segmentation experiment. I’m wondering why this is the case, considering the claim that Analogical Networks are better in the few-shot data regime.

2. It is unclear which experiments address which questions posed at the beginning of the experiments section. For example, the question “How much the performance of Analogical Networks varies with different retrieval mechanisms?” is only answered by looking through the appendix.

3. In the section for within-instance pretraining, how the encoder is trained is not well explained. The text states the encoder is pretrained using the within-instance correspondences, but figure 2 (which is meant to illustrate the pretraining) only shows the modulator. As I understand it, the encoder is not directly optimized for the retrieval objective, and gradients are propagated through the modulator to the encoder from the part correspondence objective. However, this is not clear from the figure or text.

Reproducibility:
Code snippets are provided in the appendix.

For novelty see strengths and weaknesses.


**Details Of Ethics Concerns:**

No ethical concerns.

**Strength And Weaknesses:**

Strengths:
1. Overall the proposed approach is interesting and, to my knowledge, novel. The problem of few-shot segmentation is decently motivated given that dense annotations are time consuming to obtain at scale.

2. The ablations are well thought out and provide interesting empirical insights such as how parametric methods compare to a memory-based approach for 3D object parsing.

3. The within-instance pretraining using point cloud augmentation is a clever solution to the non-differentiable aspect of the retrieval mechanism.

4. The method outperforms 3D-DETR and Prototypical Networks for 3D part segmentation.

Weaknesses:

1. The baselines compared against seem relatively weak. 3D-DETR is designed for detecting whole objects and Prototypical Networks are not designed for 3D object detection. It would be informative to compare against baselines meant for 3D part segmentation such as PartNet[1] and PriorNet[2] or explain why these methods are not suitable for comparison.

2. Qualitative results are only shown for the base classes. Given that the goal is few-shot transfer, it would be better to see qualitative results for the few-shot classes (washer, fridge, bed, table).

3. Discussion of 3D object detection in the related works would be helpful to situate the work in context. The paper states that works thus far have used separate networks to parse each semantic category, but does not explain or show experimental results to demonstrate why this approach is worse.

4. The motivation for part segmentation given in the abstract is dubious. The paper states, "Fine-grain visual parsing is necessary for ... action recognition". However, most state-of-the-art action recognition methods work directly from pixels and do not parse objects into fine-grain parts.

[1] Yu, Fenggen, et al. "Partnet: A recursive part decomposition network for fine-grained and hierarchical shape segmentation." Proceedings of the IEEE/CVF Conference on Computer Vision and Pattern Recognition. 2019.

[2] Wang, Xiaogang, et al. "Learning fine-grained segmentation of 3d shapes without part labels." Proceedings of the IEEE/CVF Conference on Computer Vision and Pattern Recognition. 2021.



**Summary Of The Paper:**

The paper proposes Analogical Networks, a method for segmenting 3D parts of objects which excel in the few-shot setting. Specifically, the method retrieves examples from a memory bank closest to the query object, then assigns correspondences between parts of the query object and those of the examples. The approach outperforms existing 3D object segmentation and meta-learning methods for few-shot 3D part segmentation. Additionally, the paper shows that part correspondences for novel classes emerge from training without part specific labels.

**Summary Of The Review:**

My initial recommendation is marginally below the acceptance threshold. Overall casting the problem of parsing 3D objects as analogical correspondence is a novel and interesting idea. However, the paper does not compare with other 3D part segmentation methods or clearly place their approach in the context of part segmentation papers. Additionally, the insights in the experiments section are difficult to follow. The empirical questions posed at the beginning of the section are interesting, however it is not clear where each one is explicitly answered. With improved clarity in the experiments section and more complete comparison to existing work I would be happy to improve my score.

---

> ### Author Response · Authors · 2022-11-13
> **Response to Reviewer MKmk (Part 1)**
>
> Thank you for your constructive review and helpful suggestions! We try to address your concerns below. We have updated our draft and highlighted the major edits in the main paper with orange text.
>
>
> >Q1: The baselines compared against seem relatively weak. 3D-DETR is designed for detecting whole objects and Prototypical Networks are not designed for 3D object detection. It would be informative to compare against baselines meant for 3D part segmentation such as PartNet[1] and PriorNet[2] or explain why these methods are not suitable for comparison.
>
> A1: Following your suggestion to compare against baselines specifically designed for part segmentation, we **added Table 4** in our Appendix that evaluates our model on instance segmentation with labels on PartNet. Since Analogical Networks semantic is already trained on PartNet, we did not need to re-train our model. This allows us to **compare with very recent approaches [5,6]**. Our model largely **outperforms all competitors (15.2% average boost over [5], 22.5% boost over [6]), using a single model**, i.e. without training on each class and level separately. Also note that our model is at disadvantage because it is tested without fine-tuning on novel classes (Bed, Dishwasher, Fridge and Table) and as a result  has never seen any examples of these classes at training time, apart from in-context usage of labeled memories at test time. In contrast, all other approaches in Table 4 use all available training data for each class. Still, we can see that it largely outperforms other competitors in three out of four novel classes.
>
> Additionally, we **trained the PartNet model of [3] on our setup** for multiple classes and segmentation levels (see **revised Table 1**). Same as in 3D-DETR, we provide the model with a level id embedding. We find that its performance is significantly worse than all our variants and other baselines, specifically **15% worse than Analogical Networks in many-shot and 27% in 1-shot**.
>
> Out of the two baselines that you suggested, PriorNet [2] tests on a custom dataset and does not have publicly available code and PartNet [1] tests on ShapeNet part, with different semantic labels. While we expect our model to work on that dataset as well, we need to re-train so as to fairly compare against [1]. This is why we preferred the comparison of Table 4, because we did not have to re-train our model and also allows us to compare against more recent (2021 and 2022) approaches, while [1] is a 2019 paper. We hope that the additional comparisons we provided above helped in clarifying the concerns about task-specific baselines.
>
> Lastly, to the best of our knowledge, most approaches that are specifically designed for part segmentation are trained per category, e.g. [1,3,4], and cannot adapt few-shot or handle multiple classes and segmentation levels with a single model. To this end, we compared Analogical networks to baselines that have this capability. We built upon Detection Transformers, designed originally to detect objects in 2D or 3D scenes, and we repurposed the model to detect parts by training it on the part label space: for us, parts are to objects as objects are to scenes. Both 3D-DETR and Analogical Networks contextualize their queries with the input, allowing them to adapt to the current observation. This dynamic behavior is not an inherent component of previous approaches [1-4].
>
> ***
>
> >Q2: Qualitative results are only shown for the base classes. Given that the goal is few-shot transfer, it would be better to see qualitative results for the few-shot classes (washer, fridge, bed, table).
>
> A2: Thank you for your suggestion. We have **added more qualitative results for novel classes in Figure 8** of the Appendix, where we can see that Analogical Networks generalize and **preserve part correspondences on novel categories** even without any fine-tuning by using memories for in-context prediction.
>
> ***

---

> > ### Author Response · Authors · 2022-11-13
> > **Response to Reviewer MKmk (Part 2)**
> >
> > >Q3: Discussion of 3D object detection in the related works would be helpful to situate the work in context.
> >
> > A3: Following your suggestion, we have **added the following paragraph about 3d object detection and part segmentation in our related work section**: “3D instance segmentation has been traditionally approached as a clustering problem (Chen et al., 2009; Sidi et al., 2011). Point-based methods learn either translation vectors mapping every point to its instance’s center (Jiang et al., 2020; Chen et al., 2021; Vu et al., 2022) or similarities across points (Wang et al., 2018a; Zhang & Wonka, 2021), followed by one or more stages of clustering. Similarly, (Wang et al., 2021; Jones et al., 2022) oversegment the point cloud into small regions and then merge them into parts. Yu et al. (2019) recursively decompose a point cloud into segments of finer resolution. (Mo et al., 2019; Sun et al., 2022) learn representative vectors that form clusters by voting for each point. However, these approaches usually assume a fixed label space and need to train a separate model for each sub-task. In contrast, we employ Detection Transformers (Carion et al., 2020) for instance segmentation by repurposing the query vectors to act as representative vectors. We extend this set of queries with memory-initialized queries, enabling in-context reasoning. This allows us to train one model across all categories. As our results show, in absence of such memory contextualization, training one model across multiple categories gives inferior results (Table 7).”.
> >
> > ***
> >
> > >Q4: The paper states that works thus far have used separate networks to parse each semantic category, but does not explain or show experimental results to demonstrate why this approach is worse.
> >
> > A4: Thank you for pointing this out. We now **quantify this claim in the Section 6.6** of the Appendix. We train PartNet [3], 3D-DETR and Analogical Networks under three setups: i) only on “Chair”, which is the most common category (see Table 6, approximately half of our base category samples fall into that category), ii) only on “Faucet”, which is a class with relatively few examples (Table 6), iii) on all classes. We test the performance of each variant on the two selected classes as well as the four novel classes we use in our paper. **Our results are summarized in Table 7**. Our conclusions are: (i) **Models trained on a single category fail to generalize to other classes even after fine-tuning**, despite their strong performance in the training class. (ii) **Analogical Networks trained on all classes outperform category-specific models on all tested setups. This is not true for the other baselines**. Both PartNet and 3D-DETR are better when trained on a single class and tested on the same class than when trained across multiple classes. Analogical Networks generalize better with more diverse data, as the model carries out an inference in context of the relevant memories.
> >
> > ***
> >
> > >Q5: The motivation for part segmentation given in the abstract is dubious. The paper states, "Fine-grain visual parsing is necessary for ... action recognition". However, most state-of-the-art action recognition methods work directly from pixels and do not parse objects into fine-grain parts.
> >
> > A5: We agree with this comment. We meant to say that part-awareness is useful for interaction with the physical world. We **revised that part in the abstract**: ”ine-grained 3D visual parsing, although useful for scene understanding and interaction with the physical world, remains underexplored.”
> >
> > ***
> >
> > >Q6: For the emergent part-correspondences experiment it appears that 3D-DETR is comparable in performance to Analogical Networks at 1-shot and 5-shot, but significantly lower at many-shot. The trend is the opposite for the part segmentation experiment. I’m wondering why this is the case, considering the claim that Analogical Networks are better in the few-shot data regime.
> >
> > A6: Thanks a lot for this comment, we discovered an issue with the dataloader that was only affecting the part semantic related experiments. We have resolved this and after performing a hyper-parameter search, we have updated Table 2 and Table 4. We observe now that the trend of performance is similar to table 1 (ie. baseline and Analogical Net have comparable many-shot performance and also Analogical Networks have better few-shot performance). Most relevant to our conclusions is the metric PQ that evaluates instance segmentation, whereas mIOU evaluates only the semantic segmentation.

---

> > > ### Author Response · Authors · 2022-11-13
> > > **Response to Reviewer MKmk (Part 3)**
> > >
> > >
> > > >Q7: It is unclear which experiments address which questions posed at the beginning of the experiments section. For example, the question “How much the performance of Analogical Networks varies with different retrieval mechanisms?” is only answered by looking through the appendix.
> > >
> > > A7: We apologize for this confusion. We **restructured our experimental section** so that the questions and answers are consistent. Specifically, we added indicators to where each question is answered in the main text and added references to our appendix for all other experiments/discussion.
> > >
> > > ***
> > >
> > > >Q8: In the section for within-instance pretraining, how the encoder is trained is not well explained. The text states the encoder is pretrained using the within-instance correspondences, but figure 2 (which is meant to illustrate the pretraining) only shows the modulator. As I understand it, the encoder is not directly optimized for the retrieval objective, and gradients are propagated through the modulator to the encoder from the part correspondence objective. However, this is not clear from the figure or text.
> > >
> > > A8: Your understanding is correct. The **encoder and the modulator form an end-to-end differentiable computational graph**: the gradients of the part segmentation objective flow from the modulator to the encoder.
> > > The retriever uses a separate encoder sub-network, with similar architecture but different weights. The encoder used for retrieval is not end-to-end trained.
> > > **We revised the figures and text to better explain this part** and now we show two different encoder networks, each with different color.
> > >
> > > Additionally, we edited the last paragraph of Section 3 to show the connection of the encoder and the retriever: “After within-instance pre-training (Algorithm 1), we maintain two copies of the encoder weights: one copy acts as the encoder of the retriever and is frozen. The other copy is used as the encoder for modulation and is further trained cross-instance, i.e. with the modulating memory being sampled from the top-k retrieved memories.” We have also added the pseudo-code in the appendix to describe the within-instance and cross-instance training separately.
> > >
> > > ***
> > >
> > > Thanks again for your constructive comments. We hope that our responses are helpful in addressing your concerns. Please don't hesitate to let us know if there are any additional questions.
> > >
> > > References:
> > >
> > > [1] Yu et al. 2019, “Partnet: A recursive part decomposition network for fine-grained and hierarchical shape segmentation”
> > >
> > > [2] Wang et al. 2021, “Learning fine-grained segmentation of 3d shapes without part labels”
> > >
> > > [3] Mo et al. 2019, “PartNet: A large-scale benchmark for fine-grained and hierarchical part-level 3D object understanding”
> > >
> > > [4] Wang et al. 2018, “SGPN: Similarity group proposal network for 3d point cloud instance segmentation”
> > >
> > > [5] Sun et al. 2022, “Semantic segmentation-assisted instance feature fusion for
> > > multi-level 3d part instance segmentation”
> > >
> > > [6] Zhang et al. 2021, “Point Cloud Instance Segmentation using Probabilistic Embeddings”
> > >
> > > [7] Chan et al. 2022, “Data Distributional Properties Drive Emergent In-Context Learning in Transformers”

---

> > > > ### Comment · Reviewer_MKmk · 2022-11-18
> > > > **Concerns Addressed**
> > > >
> > > > I thank the authors for such a detailed and thorough response. My concerns about empirical comparison to related work and clarity have been addressed.

---

> > > > > ### Author Response · Authors · 2022-11-18
> > > > > **Thank you and Response Update**
> > > > >
> > > > > Thanks again for your valuable feedback. Upon further investigation, we have updated our response to Q6, kindly refer to our updated reply above.

---

### Official Review · Reviewer_edrP · 2022-10-28

**Confidence:** 4
**Correctness:** 3
**Technical Novelty And Significance:** 3
**Empirical Novelty And Significance:** 3
**Recommendation:** 6

**Clarity, Quality, Novelty And Reproducibility:**

- This work lacks a detailed description of the retrieval section. I'm not sure if the retrieval task is only applicable to datasets with very different objects like PartNet dataset.

- Could the authors provide more experiments on different datasets, if more different datasets are easy to access. Could you please show the retrieval ability of your approach as a table? Besides, I am not sure that the inner product is the most correct way. Could you compare it with several common distance formulas?



**Strength And Weaknesses:**

- The main contributions are clearly stated.
- Appendices provide useful, supplementary information.


**Summary Of The Paper:**

The paper proposes a method to cast fine-grained visual parsing into analogical inference instead of previous methods that mapping input scenes to part labels. The main idea is to retrieve the new input point cloud feature embedding from memories of labelled feature embeddings. Then the several results with the highest similarity are retrieved are used as prior knowledge for subsequent tasks. Empirical results show that this approach achieves state-of-the-art results the PartNet dataset.

**Summary Of The Review:**

I recommend this paper towards acceptance. Using retrieval from memory is a simple and effective method. And it can inspire future related work.

---

> ### Author Response · Authors · 2022-11-13
> **Response to Reviewer edrP**
>
> Thank you for your feedback. Please see our responses to your concerns below. To address the reviewer’s concerns, we have updated our draft and highlighted the major edits in the main paper with orange text.
>
>
> >Q1: This work lacks a detailed description of the retrieval section.
>
> A1: We have **added** the following **description about the retriever in Section 3** of the main paper: “”Our model has two encoders, one used for retrieval (shown in green in Figure 2), and one for modulation (blue in Figure 2). Both encoders have a PointNet++ backbone (Qi et al., 2017).
> The encoder used in retrieval first extracts features for the input point cloud and each memory point cloud and summarizes each memory and input feature cloud into an 1D vector using average pooling. This encoder is trained with within-instance correspondence pre-training as explained later. The encoder used in modulation encodes the input scene S and each retrieved memory scene M to a set of 3D point features. 3D positional encodings are then added to the 3D point features. Each labelled part p of M is then encoded into a 1D feature vector by average pooling of its point features. This encoder is trained end-to-end with the modulator, i.e. gradients of the part segmentation objectives back-propagate to the encoder.
> Retriever: Given the memory and input normalized 1D encodings, the top-k memories are retrieved by computing an inner product between the input point cloud feature and memory features.”
>
> Additionally, we **edited the last paragraph of Section 3**: “After within-instance pre-training (Algorithm 1), we maintain two copies of the encoder weights: one copy acts as the encoder of the retriever and is frozen. The other copy is used as the encoder for modulation and is further trained cross-instance, i.e. with the modulating memory being sampled from the top-k retrieved memories.”
>
> Lastly, we **edited Figure 2** to better reflect the detailed description as mentioned above.
>
> ***
>
> >Q2: I'm not sure if the retrieval task is only applicable to datasets with very different objects like the PartNet dataset.
>
> A2: The retriever assigns high scores to objects that have similar global structure. We showcase this qualitatively in Appendix **Figures 4 and 5 for PartNet dataset and Figure 14 for ScanObjectNN dataset**, where we visualize the top-4 most similar retrieved memories for a given point cloud. Note that for different instances of the same class category we retrieve very different memories which demonstrates the fine-grained similarity between input and memories that the retriever captures. In other words, even within the same semantic category we are able to retrieve the most relevant examples. This **generalizes to novel classes as well** (see table, refrigerator and bed in Figure 5), without any finetuning of the retriever.
>
> ***
>
> >Q3: Could you please show the retrieval ability of your approach as a table? Besides, I am not sure that the inner product is the most correct way. Could you compare it with several common distance formulas?
>
> A3: Thank you for your suggestion. Since we have no direct way to quantify the retriever’s performance, we ablate the effect of different retriever distance choices on the 5-shot ARI metric. We **extend Table 5 in the Appendix to include results with different distance metrics**, such as L1, cosine and Chamfer distance. Note that we use normalized embeddings to calculate nearest neighbors, which makes cosine and L2 equivalent in terms of ranking. We find that **using either cosine or L1 metric has no statistically significant effect on the performance**. This is probably because we do not optimize for the retrieval task explicitly, thus the feature space is not regularized for a specific distance metric. However, both choices are better than Chamfer distance which operates on raw point clouds, rather than features.
>
> ***
>
> >Q4: Could the authors provide more experiments on different datasets, if more different datasets are easy to access.
>
> A4: We further **evaluate Analogical Networks on ScanObjectNN** [1], a dataset that contains noisy and incomplete point clouds of indoor objects. We observe that **both the retriever (Figure 14) and the modulator (Figures 12, 13) can generalize to this new setup**. Quantitatively, we show both many-shot and few-shot results in **Table 8**, where Analogical Networks outperform the parametric 3D-DETR and PartNet baselines, **by 1.2% and 17.5% respectively in many-shot, and by 2.2% and 22.4% in few-shot without fine-tuning**.
>
>
> ***
>
> Thanks again for your comments. We hope that our responses are helpful in answering your questions. Please let us know if there are any additional questions.
>
>
> References:
>
> [1] Uy et al. 2019, “Revisiting Point Cloud Classification: A New Benchmark Dataset and Classification Model on Real-World Data”

---

### Author Response · Authors · 2022-11-13
**Summary of Revisions**

We thank all the reviewers for their insightful and constructive feedback.

We have revised and updated our paper and appendix on openreview. Here, we summarize the added updates in the revised draft:
- Revised the abstract, contribution (with the definition of "task") and added related work for 3D Instance Segmentation
- Retriever details in section 3 (with updates in Figure 2 and 3)
- Comparison with a new baseline for instance segmentation (Table 1, 3, 7) that was proposed in the PartNet paper [1]
- Updated Table 2 after resolving a dataloader issue that was affecting part semantic models
- Appendix: Algorithm for within instance pre-training
- Appendix: Part instance segmentation AP50 comparison with 4 other baseline methods (Table 4)
- Appendix: Ablation for Retriever with different distance metric to calculate similarity (Table 5)
- Appendix: Comparison of single-category trained and multi-category trained models (Table 7) that highlights training across multiple categories is advantageous
- Appendix Section 6.5: Qualitative top 4 retriever results
- Appendix Section 6.7: Few-shot and Many-shot performance on a new dataset ScanObjectNN [2]
- Appendix: Qualitative retriever results for PartNet (Figure 4, 5) and ScanObjectNN (Figure 14) dataset
- Appendix: Qualitative parsing results for Novel classes in PartNet dataset (Figure 8) and Base and Novel classes for ScanObjectNN (Figure 12, 13)

We highlight the major updates in our revised version of the main paper with orange text.

[1] Mo et al. 2019, “PartNet: A large-scale benchmark for fine-grained and hierarchical part-level 3D object understanding”

[2] Uy et al. 2019, “Revisiting Point Cloud Classification: A New Benchmark Dataset and Classification Model on Real-World Data”

---

> ### Author Response · Authors · 2022-11-18
> **Update ( Summary of Revisions)**
>
> We have further revised our draft to improve clarity and updated experiment sections based on reviewers' feedback.

---

### Decision · Program_Chairs · 2023-01-20

**Decision:**

Accept: poster

**Justification For Why Not Higher Score:**

I am recommending an accept given the reviews, rebuttal and comments by the reviewers. Although the work has clearly been well received by the reviewers, they are not overly excited with the contributions. The presented method has limitations such as the retriever not being differentiable and not scalable. I think the reviewers would have been a lot more excited if the work even partially addressed at least some of these limitations.

I am not an expert in this area and to some extent am leaning on the reviewers expertise, and as a result, I am recommending a poster accept.

**Justification For Why Not Lower Score:**

N/A

**Metareview: Summary, Strengths And Weaknesses:**

This paper presents a method to segment 3D object parts in a few shot manner. It uses retrieval and correspondence assignment to label object parts given just a few examples. The paper presents results that outperform past works for few shot 3D segmentation. Four reviewers provided reviews for this paper. They found the approach to be novel and interesting, the ablations to be thorough and the gains of the proposed method to be significant over prior work. Some concerns included the need for stronger and recent baselines as comparisons, a need for more qualitative results and a rewrite of some sections of the paper such as the abstract. The authors have done a commendable job in their rebuttal and have addressed most concerns thoroughly including providing new quantitative comparisons. For the most part, reviewers were happy with the rebuttal and edits to the paper and two reviewers increased their scores as a result. The paper now has three accept scores (with two scores of 8) and one borderline score. Given the reviews, revisions and feedback from the reviewers, I am recommending an accept.

**Note From Pc:**

if the above contains the word "oral" or "spotlight" please see: "oral" presentation means -> notable-top-5% and "spotlight" means -> notable-top-25%. As stated in our emails, we are disassociating presentation type from AC recommendations

**Summary Of Ac-Reviewer Meeting:**

N/A